# COOPERATIVE MULTI-AGENT PLANNING WITH ADAPTIVE SKILL SYNTHESIS

## ABSTRACT

Cooperative multi-agent reinforcement learning (MARL) struggles with sample efficiency, interpretability, and generalization. While Large Language Models (LLMs) offer powerful planning capabilities, their application has been hampered by a reliance on text-only inputs and a failure to handle the non-Markovian, partially observable nature of multi-agent tasks. We introduce COMPASS, a multi-agent framework that overcomes these limitations by integrating Vision-Language Models (VLMs) for decentralized, closed-loop decision-making. COMPASS dynamically generates and refines interpretable, code-based strategies stored in a skill library that is bootstrapped from expert demonstrations. To ensure robust coordination, it propagates entity information through a structured multi-hop communication protocol, allowing teams to build a coherent understanding from partial observations. Evaluated on the challenging SMACv2 benchmark, COMPASS significantly outperforms state-of-the-art MARL baselines. Notably, in the symmetric Protoss 5v5 task, COMPASS achieved a 57% win rate, a 30 percentage point advantage over QMIX (27%). Project page can be found at https://stellar-entremet-1720bb.netlify.app/.

## 1 INTRODUCTION

A major long-term goal for the field of cooperative multi-agent systems (MAS), e.g. multi-robot control Gao et al. (2024); Feng et al. (2024), power management Monroc et al. (2024), and multi-agent games Samvelyan et al. (2019); Kurach et al. (2020), is to build a protocol of collaboration among the agents. Multi-agent reinforcement learning (MARL), proven as an advanced paradigm of distributed artificial intelligence (AI), holds promise for discovering collective behavior from interactions. One line of works follows centralized training decentralized execution (CTDE) paradigm Rashid et al. (2020); Yu et al. (2022); Liu et al. (2024); Lowe et al. (2017); Zhu et al. (2025); Li et al. (2024b). CTDE assumes a central controller that exploits global information, while the individual policies are designed to allow for decentralized execution. However, in many real-world scenarios, the central controller becomes unfeasible due to the communication overhead that exponentially scales with the number of agents, thereby compromising the scalability of MAS. In contrast, the decentralized training decentralized execution paradigm (DTDE) discards this assumption and is scalable to large-scale systems Su & Lu (2024); Zhang et al. (2018); Ma et al. (2024a). However, DTDE requires complicated learning and planning under uncertainty, as partial observability magnifies the discrepancy between each agent's local observation and global information. Although much progress has been made, MARL suffers from compromised sample efficiency, interpretability, and transferability.

The emergence of Large Language Models (LLMs) has revitalized this field. LLM-based multi-agents have been proposed to leverage their remarkable capacity to perform task-oriented collective behaviors Mandi et al. (2024); Zhang et al. (2024); Gong et al. (2024); Zhang et al. (2025); Nayak et al. (2025). The LLMs are used for high-level planning to generate centralized Gong et al. (2024); Nayak et al. (2025); Deng et al. (2024) or decentralized plans Mandi et al. (2024); Zhang et al. (2024), often adopting a hierarchical decision-making structure in conjunction with a pre-defined low-level controller. While these methods have succeeded in a set of multi-agent problems including Overcooked-AI Carroll et al. (2019), SMAC Samvelyan et al. (2019), and VirtualHome Puig et al. (2018), heavy reliance on text-based observation prevents them from learning from multi-modal

information. Moreover, they ignore the non-Markovian nature of MAS, where learning and planning necessitate a decentralized closed-loop solution.

In this paper, we mitigate the existing research limitations and advance general decision-making for cooperative multi-agent systems. At a high level, COMPASS combines a vision language models (VLMs)-based planner with a dynamic skill library for storing and retrieving complex behaviors, along with a structured communication protocol. A diagram of COMPASS is provided in Figure 1. Inspired by Cradle Tan et al. (2024), the VLM-based planner perceives the visual and textual observation and suggests the most suitable executable code from the skill library. We adopt the code-as-policy paradigm Wang et al. (2024d) instead of task-specific primitive actions, as it constrains generalizability and fails to fully leverage foundation models' extensive world knowledge and sophisticated reasoning capabilities.

Traditional open-loop methods struggle to produce effective plans that adapt to dynamics in stochastic, partially observable environments. To address this challenge, the VLM-based planer attempts to solve challenging and ambiguous final tasks, such as "Defeat all enemy units in the StarCraft multi-agent combat scenario while coordinating with allied units", by progressively proposing a sequence of clear, manageable sub-tasks while incorporating environmental feedback and task progress. COMPASS generates Python scripts through LLMs as semantic-level skills to accomplish sub-tasks, incrementally building a skill library throughout the task progress. Each skill is indexed through its documentation embeddings, enabling retrieval based on task-skill relevance. However, developing the skill library from scratch requires extensive exploration to discover viable strategies. In contrast, we pre-collect demonstration videos and introduce a "warm start" by initializing the skill library with strategies derived from the expert-level dataset.

Moreover, building autonomous agents to cooperate in completing tasks under partial observation requires an efficient communication protocol. However, naive communication leads to the risks of hallucination caused by meaningless chatter between agents Li et al. (2024a). Inspired by entity-based MARL Iqbal et al. (2021); Ding et al. (2023), we present a structured communication protocol to formulate the communication among agents along with a global memory that allows all agents to retrieve. The protocol incorporates a multi-hop propagation mechanism, enabling agents to infer information about entities beyond their field of view through information shared by teammates. Similar to previous approaches, each agent maintains a local memory to preserve current and historical experiences.

Empirically, COMPASS demonstrates effective adaptation and skill synthesis in cooperative multi-agent scenarios. Through its dynamic skill library, it creates reusable and interpretable code-based behaviors that evolve during task execution. We evaluate COMPASS systematically in the improved StarCraft Multi-Agent Challenge (SMACv2) using both open-source (Qwen2-VL-72B Wang et al. (2024c)) and closed-source (GPT-4o-mini[1], Claude-3-Haiku[2]) VLMs. COMPASS achieves strong results in Protoss scenarios with a 57% win rate, substantially outperforming state-of-the-art MARL algorithms, including QMIX Rashid et al. (2020), MAPPO Yu et al. (2022), HAPPO Kuba et al. (2022), and HASAC Liu et al. (2024). COMPASS maintains moderate performance in Terran scenarios and handles asymmetric settings effectively, though showing limited success in Zerg task. We evaluate the contribution of individual COMPASS components to its overall performance. We further demonstrate COMPASS's ability to bootstrap effective strategies from expert demonstrations.

## 2 RELATED WORK

### 2.1 AGENTS IN THE STARCRAFT MULTI-AGENT CHALLENGE

SMAC Samvelyan et al. (2019), a predominant cooperative MARL benchmark based on the StarCraft II real-time strategy game Vinyals et al. (2017), focuses on decentralized micromanagement scenarios where each unit operates under decentralized execution with partial observability to defeat enemy units controlled by Starcraft II's built-in AI opponent. Previous research in SMAC resort to MARL which can be divided into two categories: 1) Online MARL: One line of representative research is value decomposition (VD) Rashid et al. (2020); Wang et al. (2021); Son et al. (2019), which

---

[1] https://platform.openai.com/docs/models#gpt-4o-mini
[2] https://www.anthropic.com/news/claude-3-haiku

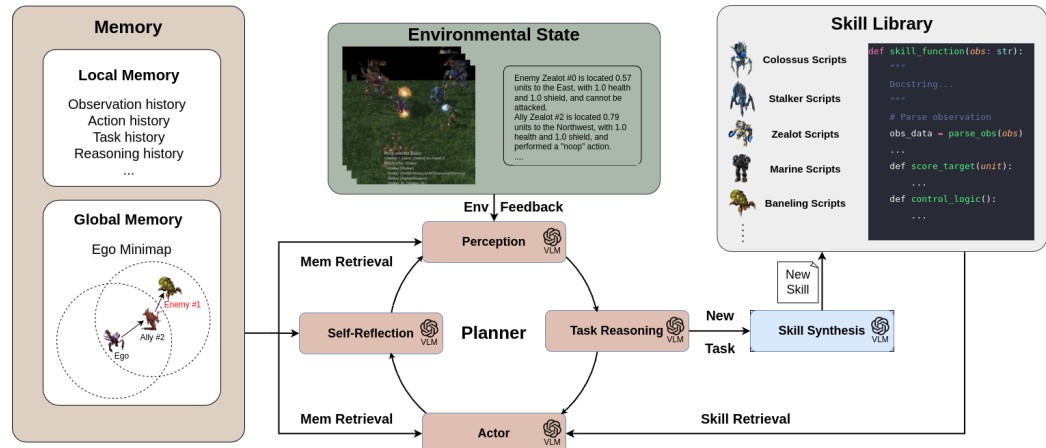

Figure 1: Overview of the COMPASS architecture, a novel framework that advances cooperative multi-agent decision-making through three synergistic components: (1) A VLM-based closed-loop planner that enables decentralized control by continuously processing multi-modal feedback and adapting strategies, addressing the non-Markovian challenge of multi-agent systems; (2) A dynamic skill synthesis mechanism that combines demonstration bootstrapping with incremental skill generation, improving sample efficiency and interpretability; and (3) A structured communication protocol that facilitates efficient information sharing through entity-based multi-hop propagation, enhancing cooperative perception under partial observability.

decomposes the centralized action-value function into individual utility functions. On the other hand, multi-agent policy gradient (MAPG) methods Yu et al. (2022); Kuba et al. (2022); Liu et al. (2024); Li et al. (2023); Wen et al. (2022); Hu et al. (2024); Na et al. (2024) extend single-agent policy gradient algorithm to multi-agent with coordination modeling. Researches such as MAPPO Yu et al. (2022), HAPPO Kuba et al. (2022), and HASAC Liu et al. (2024) combine trust region and maximum entropy with MARL in a non-trivial way respectively. To encourage coordination, communication methods Hu et al. (2024); Lo et al. (2024), sequential modeling methods Li et al. (2023); Wen et al. (2022), and cooperative exploration methods Mahajan et al. (2019); Na et al. (2024) have been proposed. 2) Offline MARL: Recent efforts such as MADT Meng et al. (2023), ODIS Zhang et al. (2022), and MADiff Zhu et al. (2025) leverage data-driven training via pre-collected offline datasets to enhance policy training efficiency. However, the near-optimal performance of these existing approaches on SMAC highlights the benchmark's limited stochasticity and partial observability. To address these limitations, SMACv2 Ellis et al. (2024) introduces more complexity to necessitate decentralized closed-loop control policies. There have been some recent attempts Li et al. (2024b); McClellan et al. (2024); Formanek et al. (2023); Li et al. (2024c) to evaluate MARL algorithms on SMACv2, and the results confirm the complexity. However, current learning-based multi-agent methods are computationally inefficient and non-interpretable. In the quest to find methods that are sample-efficient and interpretable, LLM-SMAC Deng et al. (2024) leverage LLMs to generate centralized decision tree code under global information in an open-loop framework. Unlike prior works, COMPASS integrates a Vision-Language Model (VLM) with each agent in a decentralized closed-loop manner under partial observability, improving both real-world applicability and scalability.

## 2.2 LLM-BASED MULTI-AGENT SYSTEM

Based on the inspiring capabilities of LLMs, such as zero-shot planning and complex reasoning Kojima et al. (2024); Zhao et al. (2024); Wei et al. (2024); Besta et al. (2024), embodied single-agent researches have demonstrated the effectiveness of LLMs in solving complex long-horizon tasks Wang et al. (2024b); Yao et al. (2023); Shinn et al. (2023); brian ichter et al. (2022); Ma et al. (2024b); Wang et al. (2024d); Tan et al. (2024). Despite significant advances in single-agent applications, developing real-world multi-agent systems with foundation models remains challenging, primarily due to the nature of decentralized control under partial observability in multi-agent settings

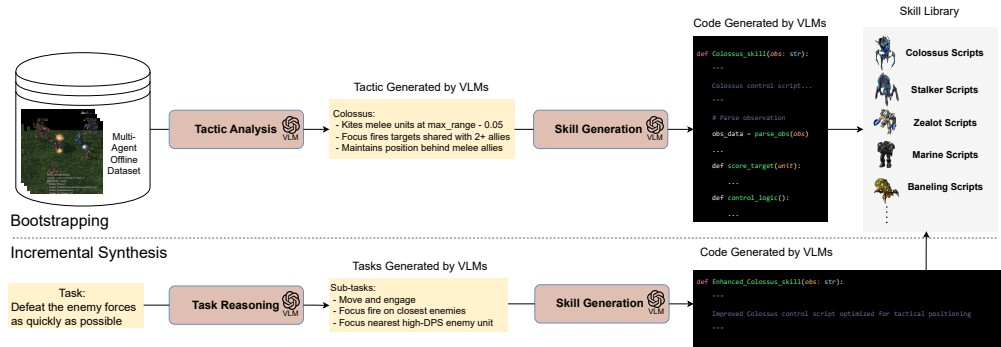

Figure 2: Overview of Adaptive Skill Synthesis. VLMs perform (Top) Bootstrapping by analyzing offline data for initial Tactic Analysis and Skill Generation into a Skill Library. (Bottom) Incremental Synthesis uses Task Reasoning to dynamically generate or enhance code-based skills, evolving the library for new tasks. The skills follow a structured decision-making pipeline with two core components: *score_target(unit)* for dynamic target prioritization and *control_logic()* for coordinating behavior. Textual observations are parsed into structured data (obs_data), mapping raw text to attributes, e.g., "Can move North: yes" to can_move='north': True.

Pajarinen & Peltonen (2011). Most prior efforts Mandi et al. (2024); Zhang et al. (2024; 2025); Nayak et al. (2025); Gong et al. (2024) leverage a hierarchical framework with components like perception, communication, planning, execution, and memory to build multi-agent systems with collective behaviors. These approaches can be roughly classified into two groups. 1) Centralized plan: MindAgent Gong et al. (2024) adopts a centralized planning scheme with a pre-defined oracle in a fully observable multi-agent game. LLaMAR Nayak et al. (2025) employs LLMs to manage long-horizon tasks in partially observable environments without assumptions about access to perfect low-level policies. 2) Decentralized plan: ProAgent Zhang et al. (2025) introduces Theory of Mind (ToM), enabling agents to reason about others' mental states. RoCo Mandi et al. (2024) and CoELA Zhang et al. (2024) assign separate LLMs to each embodied agent for collaboration with communication. However, RoCo and CoELA assume a skill library with a low-level heuristic controller, which is impractical in real-world applications. Moreover, RoCo's open-loop plan-and-execute paradigm fails to incorporate environmental feedback during decision-making. In contrast, our work does not assume any pre-defined low-level controller and generates code-based action through VLMs in a closed-loop manner.

## 3 PRELIMINARIES

We model a fully cooperative multi-agent game with $N$ agents as a *decentralized partially observable Markov decision process* (Dec-POMDP) Oliehoek & Amato (2016), which is formally defined as a tuple $\mathcal{G} = (\mathcal{N}, \mathcal{S}, \mathcal{O}, \mathbb{O}, \mathcal{B}, \mathcal{A}, \mathcal{T}, \Omega, R, \gamma, \rho_0)$. $\mathcal{N} = \{1, \ldots, N\}$ is a set of agents, $s \in \mathcal{S}$ denotes the state of the environment and $\rho_0$ is the distribution of the initial state. $\mathcal{A} = \prod_{i=1}^{N} A^i$ is the joint action space, $\mathbb{O} = \prod_{i=1}^{N} O^i$ is the set of joint observations. At time step $t$, each agent $i$ receives an individual partial observation $o_t^i \in O^i$ given by the observation function $\mathcal{O} : (a_t, s_{t+1}) \mapsto P(o_{t+1}|a_t, s_{t+1})$ where $a_t, s_{t+1}$ and $o_{t+1}$ are the joint actions, states and joint observations respectively. Each agent $i$ uses a stochastic policy $\pi^i(a_t^i|h_t^i, \omega_t^i)$ conditioned on its action-observation history $h_t^i = (o_0^i, a_0^i, \ldots, o_{t-1}^i, a_{t-1}^i)$ and a random seed $\omega_t^i \in \Omega_t$ to choose an action $a_t^i \in A^i$. A belief state $b_t$ is a probability distribution over states at time $t$, where $b_t \in \mathcal{B}$, and $\mathcal{B}$ is the space of all probability distributions over the state space. Actions $a_t$ drawn from joint policy $\pi(a_t|s_t, \omega_t)$ conditioned on state $s_t$ and joint random seed $\omega_t = (\omega_t^1, \ldots, \omega_t^N)$ change the state according to transition function $\mathcal{T} : (s_t, a_t^1, \ldots, a_t^N) \mapsto P(s_{t+1}|s_t, a_t^1, \ldots, a_t^N)$. All agents share the same reward $r_t = R(s_t, a_t^1, \ldots, a_t^N)$ based on $s_t$ and $a_t$. $\gamma$ is the discount factor for future rewards. Agents try to maximize the expected total reward, $\mathcal{J}(\pi) = \mathbb{E}_{s_0, a_0, \ldots}[\sum_{t=0}^{\infty} \gamma^t r_t]$, where $s_0 \sim \rho_0(s_0), a_t \sim \pi(a_t|s_t, \omega_t)$.

# 4 METHODS

COMPASS, illustrated in Figure 1, is a decentralized closed-loop framework for cooperative multi-agent systems that continuously incorporate environmental feedback for strategy refinement. The architecture comprises three core components: 1) a VLM-based closed-loop planner that iteratively perceives, reasons, reflects and acts to adaptively complete tasks (Sec. 4.1); 2) an adaptive skill synthesis mechanism for generating executable codes tailored to proposed sub-tasks (Sec. 4.2); and 3) a structured communication protocol that enables agents to share visible entity information under partial observability (Sec. 4.3). The pseudo-code of COMPASS is shown in Appendix.

## 4.1 VLM-BASED CLOSED-LOOP PLANNER

Inspired by recent advances in cognitive architectures for autonomous systems Tan et al. (2024), COMPASS implements a sophisticated modular planning framework that emulates key aspects of cognitive decision-making. The planner adopts a modular formulation, utilizing four specialized models: Perception, Task Reasoning, Self-Reflection, and Actor. Each model fulfills a distinct yet interconnected role in the decision-making process. The Perception model processes multi-modal inputs, integrating both visual and textual information to build comprehensive environmental understanding. The Task Reasoning model analyzes the perceived information to decompose complex objectives into manageable sub-tasks, ensuring systematic progress toward the final goal. The Self-Reflection model continuously evaluates task execution and outcome quality, enabling adaptive behavior refinement. The Actor model translates plans into actions by selecting and executing the most appropriate skills from the skill library. We next discuss the various components in detail:

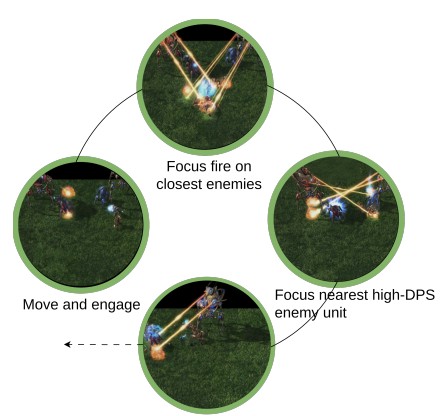

Figure 3: Visualization of COMPASS's dynamic task reasoning process in the Star-Craft Multi-Agent Challenge (SMACv2) environment. The figure demonstrates how the VLM-based planner decomposes a complex final goal ("defeat all enemy units") into a sequence of concrete, executable sub-tasks that adapt to the changing battlefield conditions. This closed-loop task decomposition enables efficient coordination among multiple agents under partial observability, as each sub-task provides clear, actionable objectives that agents can execute while maintaining overall mission alignment.

**Perception** forms the foundation of COMPASS's decision-making capabilities by enabling robust multi-modal understanding of complex environments. Solving complex real-world tasks often involves data of multiple modalities Wang et al. (2024a), each contributing unique and complementary information for decision-making. We leverage the VLMs' ability to fuse and analyze a broader spectrum of data, including text- and image-based environment feedback, to enable agents to sense the surrounding environment. The system's perception mechanism operates at two levels: direct observation processing and collaborative information synthesis. At the direct level, VLMs process raw inputs to extract meaningful features and relationships from both visual and textual data. At the collaborative level, COMPASS addresses the inherent challenge of partial observability in multi-agent systems through an innovative multi-hop communication protocol (detailed in the Structured Communication Protocol section) that enables agents to construct a more holistic understanding of their environment by sharing and aggregating observations. This dual-level perception architecture ensures that each agent maintains both detailed local awareness and broader contextual understanding, essential for effective decision-making in complex cooperative tasks.

**Task Reasoning** enables COMPASS to systematically approach complex cooperative challenges through collective task decomposition. Given a simple general final task in the cooperative multi-agent setting, e.g., "defeat all enemy units", in order to complete the task more efficiently, agents are required to decompose it into multiple sub-tasks and figure out the right one to focus on, while considering alignment among others (See Figure 3). COMPASS harnesses the power of VLMs to

analyze high-level task instructions in conjunction with environmental feedback and team member objectives to generate tractable sub-tasks that collectively advance the overall mission. As agents act under stochastic, partially observable environments, the task reasoning model continuously adapts its plans, proposing and refining sub-tasks based on emerging situations and progress assessment. This dynamic approach enables COMPASS to maintain strategic coherence while adjusting tactical decisions in response to changing circumstances.

**Actor** serves as the critical bridge between high-level reasoning and concrete action execution. Building upon recent advances in code-writing language models for embodied control Liang et al. (2023); Wang et al. (2024d), the Actor leverages the skill library by first identifying relevant skills for the proposed sub-task, then synthesizes perception and self-reflection inputs to select the optimal skill for execution. This streamlined approach ensures efficient skill selection while maintaining task alignment.

**Self-Reflection** enables COMPASS to continuously evaluate and refine its decision-making processes through systematic performance analysis (See Figure 4). COMPASS instantiates the Self-Reflection model as a VLM which takes a sequence of visual results from the last skill execution with corresponding descriptions as input to assess the quality of the decision produced by the Actor and whether the task was completed. Additionally, we also request the VLM to generate verbal self-reflections to provide valuable feedback on the completion of the task.

### 4.2 Adaptive Skill Synthesis

COMPASS employs a dynamic skill library that maintains and evolves a collection of executable behaviors. Each skill is represented as an executable Python function with comprehensive documentation describing its functionality and corresponding embedding that enables semantic retrieval. This skill library undergoes continuous refinement through two complementary mechanisms (Figure 2): incremental synthesis, where new skills are generated and existing ones are refined during task execution, and demonstration-based bootstrapping, which initializes the library with behaviors extracted from expert demonstrations. For further details and visualizations of example synthesized tactical behaviors, please refer to the Skill Analysis section in Appendix.

Figure 4: Dynamic skill evolution through self-reflection. COMPASS continuously refines its tactical capabilities by analyzing the performance of executed skills. Here, feedback on the 'stalker_cover' skill reveals a need for more aggression. This insight prompts the immediate generation and registration of a new, specialized skill ('Enhanced Stalker Script Type Aggressive'), expanding the agent's behavioral repertoire.

**Incremental Synthesis** With the Task Reasoning component consistently proposing sub-tasks, COMPASS first attempts to retrieve relevant skills from the library using semantic similarity between the sub-task description and skill documentation embeddings. If no suitable skill exists, or if existing skills prove inadequate, the VLM generates a new Python script specifically tailored to the sub-task.

**Bootstrapping** However, developing the skill library from scratch requires extensive interactions with environments, which potentially leads to inefficient learning in the early stages. Inspired by offline MARL approaches Meng et al. (2023); Zhang et al. (2022); Zhu et al. (2025), which leverage pre-collected datasets to enhance sample efficiency, we leverage MAPPO as the behavior policy to collect experiences, which are recorded as video sequences. The VLMs then analyze these demonstrations through a multi-stage process: first identifying key strategic patterns and behavioral primitives, then translating these patterns into executable Python functions with appropriate documentation. This initialization methodology establishes a foundational set of validated skills, substantially reducing the exploration overhead typically required for discovering effective behaviors. The resulting baseline

skill library enables efficient task execution from the onset while maintaining the flexibility to evolve through incremental synthesis.

### 4.3 STRUCTURED COMMUNICATION PROTOCOL

To facilitate effective collaboration under partial observability, recent LLM-based multi-agent work Li et al. (2024a); Zhang et al. (2024) employs conversational framework with unconstrained communication protocol. However, while natural language offers flexibility, unrestricted communication can lead to potential hallucinations caused by ambiguous or irrelevant messages between agents. Drawing from advances in structured communication frameworks Hong et al. (2024) and entity-based MARL Iqbal et al. (2021); Ding et al. (2023), COMPASS implements a hierarchical communication protocol that focuses on efficient entity-based information sharing and multi-hop propagation (Figure 5). Each agent maintains an observation buffer containing information about entities in its field of view. At each timestep, agents share their local observations, which are then aggregated into a global entity memory accessible to all. COMPASS employs a multi-hop communication mechanism to propagate information about distant entities, enabling agents to build a more holistic observation of the environment by leveraging the collective knowledge of the team.

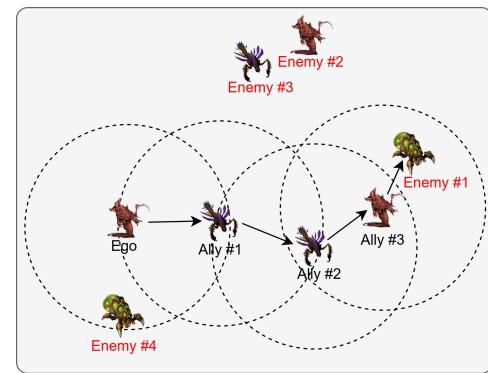

Figure 5: Illustration of COMPASS's structured multi-hop communication protocol that enables efficient information sharing under partial observability. The figure demonstrates how information about Enemy #1 propagates to the Ego agent through a chain of allied units (Ally #1, #2, #3), despite Enemy #1 being outside Ego's sight range. Each dashed circle represents an agent's local observation field, while arrows indicate the flow of entity-based information sharing. This mechanism enables agents to build a more holistic understanding of the environment by propagating critical information (e.g., enemy positions, status) through intermediate allies, effectively addressing the partial observability challenge in decentralized multi-agent systems.

## 5 EXPERIMENTS

We conducted a comprehensive experimental evaluation of COMPASS to assess its performance and capabilities in complex multi-agent scenarios. Our evaluation focused on the improved StarCraft Multi-Agent Challenge (SMACv2) Ellis et al. (2024), which provides an ideal testbed for examining cooperative behavior under partial observability and stochasticity. Through systematic experimentation, we investigated two fundamental questions: (1) How does COMPASS perform compared to state-of-the-art MARL methods? (2) What are the individual contributions of each component in COMPASS? Experiments utilize both open-source (Qwen2-VL-72B) and closed-source VLMs (GPT-4o-mini, Claude-3-Haiku), with Jina AI embeddings for skill retrieval. All results are averaged over 5 seeds to account for environmental stochasticity. Token usage is approximately 0.4 million per episode.

### 5.1 EXPERIMENTAL SETUP

**Scenarios** Our evaluation scenarios span three distinct race matchups (Protoss, Terran, and Zerg) and two categories (symmetric and asymmetric), as detailed in Appendix. The symmetric scenarios (5v5) test coordination in balanced engagements, while asymmetric scenarios (5v6) evaluate adaptation to numerical disadvantages. Each race combination presents unique tactical challenges due to different unit abilities and constraints. We followed the setting p=0 in the SMACv2 original paper (i.e., prob_obs_enemy: 0.0 in the .yaml file), meaning that only the first agent to initially spot a specific enemy unit can continue observing it, introducing the *Extended Partial Observability Challenge*, which baselines struggled with.

Table 1: Comparative performance of COMPASS (with three VLM variants: G-4o=GPT-4o-mini, C-Hk=Claude-3-Haiku, Q2-VL=Qwen2-VL-72B) and state-of-the-art MARL baselines on SMACv2. Median win rates (%) and standard deviations (subscripts) are reported across Protoss, Terran, and Zerg scenarios in symmetric (5v5) and asymmetric (5v6) categories. Results are averaged over 5 seeds. Bold values denote the best performance in each scenario.

| | QMIX | MAPPO | HAPPO | HASAC | COMPASS | | |
| --- | --- | --- | --- | --- | --- | --- | --- |
| | | | | | G-4o | C-Hk | Q2-VL |
| PROTOSS | | | | | | | |
| SYMMETRIC | $0.27_{0.03}$ | $0.32_{0.067}$ | $0.34_{0.07}$ | $0.20_{0.08}$ | $\mathbf{0.57_{0.08}}$ | $0.49_{0.06}$ | $0.45_{0.04}$ |
| ASYMMETRIC | $0.01_{0.01}$ | $0.04_{0.04}$ | $0.02_{0.03}$ | $0.01_{0.02}$ | $\mathbf{0.08_{0.04}}$ | $0.06_{0.05}$ | $0.06_{0.03}$ |
| TERRAN | | | | | | | |
| SYMMETRIC | $0.38_{0.04}$ | $0.36_{0.1}$ | $0.35_{0.1}$ | $0.29_{0.01}$ | $\mathbf{0.39_{0.01}}$ | $0.38_{0.05}$ | $0.31_{0.02}$ |
| ASYMMETRIC | $0.06_{0.02}$ | $0.07_{0.06}$ | $0.01_{0.03}$ | $0.05_{0.02}$ | $\mathbf{0.1_{0.03}}$ | $\mathbf{0.1_{0.01}}$ | $0.06_{0.03}$ |
| ZERG | | | | | | | |
| SYMMETRIC | $0.21_{0.01}$ | $\mathbf{0.27_{0.04}}$ | $0.2_{0.11}$ | $0.24_{0.07}$ | $0.16_{0.07}$ | $0.18_{0.02}$ | $0.14_{0.03}$ |
| ASYMMETRIC | $\mathbf{0.18_{0.03}}$ | $0.13_{0.09}$ | $0.09_{0.02}$ | $0.08_{0.05}$ | $0.03_{0.01}$ | $0.04_{0.01}$ | $0.02_{0.01}$ |

**Baselines** We compared COMPASS against the state-of-the-art MARL algorithms representing both value-based and policy-gradient approaches:

- Value-Based Methods: QMIX Rashid et al. (2020) uses a mixing network architecture to decompose joint action-values while maintaining monotonicity constraints.

- Policy Gradient Methods: MAPPO Yu et al. (2022) extends PPO to multi-agent settings with the CTDE paradigm. HAPPO Kuba et al. (2022) performs sequential policy updates by utilizing other agents' newest policy under the CTDE framework and provably obtains the monotonic policy improvement guarantee. HASAC Liu et al. (2024) combines the maximum entropy framework with trust region optimization to enhance exploration and coordination. MATWen et al. (2022) models the multi-agent decision process as a sequence-to-sequence generation problem with a powerful transformer architecture.

- Communication-based Methods: CommFormer Hu et al. (2024) is a method for learning optimal communication graphs in multi-agent systems using attention.

- Offline Methods: Oryx Formanek et al. is a state-of-the-art offline MARL algorithm, addressing extrapolation error and miscoordination.

- LLM-based Methods: LLM-SMAC Deng et al. (2024) generates decision tree code using Large Language Models (GPT-4o-mini) to solve the SMAC task. We modified its codebase for SMACv2, but the resulting win rates were 0% across all settings. Given this performance, we ignore this baseline in our final comparison.

**Datasets** To enable effective bootstrapping of the skill library, we constructed a comprehensive demonstration dataset capturing diverse multi-agent strategies and interactions. We employed MAPPO with original hyper-parameters as our behavior policy for data collection, leveraging its strong performance in cooperative multi-agent tasks. Our final dataset comprises over 300 complete game episodes, each recorded as a video sequence capturing the full state-action trajectory. These demonstrations span all symmetric scenario types described in Appendix.

## 5.2 MAIN RESULTS

**Performance** As shown in Table 1, COMPASS demonstrates significant performance advantages in SMACv2, particularly excelling in Protoss scenarios where it achieves a 57% win rate in symmetric engagements using GPT-4o-mini, substantially outperforming traditional approaches like QMIX (27%), MAPPO (32%), and HAPPO (34%). Due to space limitations, we have moved MAT, CommFormer, and Oryx to the appendix.

However, performance varies across race matchups. While maintaining strong results in Terran scenarios (39% win rate), COMPASS shows limited effectiveness in Zerg scenarios (16% win rate). This performance disparity can be attributed to the unique mechanics of Zerg combat units, which

Table 2: Win rates of the initialized skill library (bootstrapped from expert demonstrations) on SMACv2.

|      | PROTOSS | TERRAN | ZERG |
|------|---------|--------|------|
| 5V5  | $0.35_{0.06}$ | $0.24_{0.04}$ | $0.06_{0.01}$ |
| 5V6  | $0.04_{0.05}$ | $0.06_{0.02}$ | $0.02_{0.03}$ |

Table 3: Communication robustness under varying hop counts and packet loss rates.

| Setting | Specification | Mean Win Rate | Std Dev |
|---------|---------------|---------------|---------|
| **Hop Count** | 1-hop | 0.46 | 0.19 |
|  | 2-hop | 0.54 | 0.06 |
|  | **3-hop (Default)** | **0.57** | **0.08** |
| **Packet Loss** | 20% (0.2 fault) | 0.32 | 0.03 |
|  | 50% (0.5 fault) | 0.12 | 0.02 |
|  | 80% (0.8 fault) | 0.07 | 0.05 |
|  | 100% (No Comms) | 0.06 | 0.04 |

demand more fine-grained micromanagement due to their shorter attack ranges and reliance on swarm-based tactics.

In asymmetric scenarios (5v6), COMPASS consistently outperforms MARL baselines in Protoss and Terran matchups, demonstrating its ability to execute effective strategies despite being outnumbered. Its success in these settings suggests that COMPASS can adapt dynamically, using coordinated tactics and learned skills to counteract numerical disadvantages. The robust performance holds across different VLM implementations, with GPT-4o-mini consistently achieving the strongest results.

Moreover, COMPASS demonstrates particular advantages in scenarios with sparse reward, where traditional MARL approaches significantly underperform (near-zero win rates).

## 5.3 ABLATION STUDIES

**Skill Initialization** To evaluate the impact of our skill initialization, we analyze the performance of COMPASS using only the initialized skill library derived from expert demonstrations. The results in Table 2 demonstrate that skill initialization alone achieves non-trivial performance across different scenarios, particularly in symmetric matchups. Moreover, the gap between initialized skills and COMPASS underscores the necessity of incremental skill synthesis. A script example for skill initialization is in Appendix.

**Communication** To demonstrate the critical role of communication, we evaluated COMPASS on Protoss 5v5 under the *Extended Partial Observability* setting, using only local information without multi-hop propagation. The resulting win rate with GPT-4o-mini decreased to $0.06_{0.04}$, a significant drop from $0.57$ with full communication. This degradation occurs because the Extended Partial Observability setting restricts direct enemy visibility to the first agent that initially spots it. The VLMs generated control logic heavily relies on the presence of enemies in the local observation to determine engagement and targeting. Without communication relaying enemy positions, agents other than the discoverer cannot 'see' enemies known to teammates, even if within attack range. Consequently, their control logic frequently defaults to 'no enemy' behaviors, such as moving towards allies or executing random default actions, preventing effective target engagement and coordinated attacks, thus drastically reducing combat effectiveness and the overall win rate. We study the effect of a communication fault. As shown in the Table 3, the system degrades under moderate packet loss and suffers a drastic drop at >50% loss.

**VLM call frequency** We study the impact of VLM call frequency. As shown in the Table 4, at higher call frequencies, the performance remains similar but incurs greater cost, while lower frequencies fail to keep up with the dynamics of the battlefield.

Table 4: VLM call frequency on Protoss 5v5 (average episode length ∼60 steps).

| Frequency | Mean Win Rate | Std Dev |
|---|---|---|
| Every 10 steps | 0.56 | 0.05 |
| **Every 20 steps** | **0.57** | **0.08** |
| Every 40 steps | 0.40 | 0.08 |

**Self Reflection** In order to show the effectiveness of self-reflection, we evaluate the performance of COMPASS w/o self-reflection on protoss 5 vs 5. Removing the module leads to a drop -10% ($0.47_{0.04}$).

**Visual information** We tested visual information contribution by omitting the image inputs. Intuitively, this challenges the agents to rely solely on uni-modal textual information, potentially leading to a loss of spatial understanding. Empirical results show a 10% drop in performance when visuals are omitted. Comparing the VLM outputs shows that without visual information, understanding map boundaries relies on interpreting textual cues (e.g., 'West (unavailable movement)'), which might be derived indirectly from action availability rather than direct observation. With visual input, the VLM can directly perceive these crucial spatial details like map boundaries from the image itself. This lack of spatial awareness can result in less informed tactical decisions regarding movement and positioning.

## 6 CONCLUSION

This paper introduced COMPASS, a novel framework designed to address critical challenges in sample efficiency, interpretability, and adaptability that persist in traditional multi-agent reinforcement learning. By integrating a VLM-based closed-loop planner, a dynamic skill library bootstrapped from demonstrations, and a structured multi-hop communication protocol, COMPASS empowers agents to reason, adapt, and collaborate effectively under partial observability. Our extensive evaluations on the challenging SMACv2 benchmark confirm the superiority of this approach; notably, COMPASS achieved a 57% win rate in the symmetric Protoss scenario, outperforming a state-of-the-art QMIX baseline by 30 percentage points. While performance in micromanagement-heavy Zerg scenarios highlights an area for future improvement, our results demonstrate that COMPASS is a step toward creating more general, interpretable, and effective multi-agent systems for complex, real-world applications.

## ETHICS STATEMENT

This research was conducted in a simulated StarCraft II environment, involving no human subjects or personal data. The dataset was generated algorithmically within the simulation, mitigating privacy concerns. We acknowledge that the autonomous coordination techniques developed could be applied to technologies with real-world impact, demanding careful ethical consideration. Additionally, our framework's reliance on large Vision-Language Models introduces the risk of inherited data biases, which could lead to unintended behaviors in broader applications. We advocate for continued research in AI safety, transparency, and alignment to guide the responsible development of such systems.

## REPRODUCIBILITY STATEMENT

The complete source code for the COMPASS framework, including the implementation of the VLM-based planner, adaptive skill synthesis, and structured communication protocol, has been included in the supplementary materials. This package contains all necessary components to replicate the experiments presented in the paper, including the evaluation scripts for the SMACv2 benchmark and the code for the baseline MARL algorithms. The experimental setup is described in detail in Section 5. Further implementation details and the core algorithm's pseudocode are provided in the Appendix to facilitate a thorough understanding and verification of our methodology and results.

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

## A  PSEUDOCODE

The pseudo-code of the COMPASS algorithm is shown in Pseudocode 1.

## B  IMPLEMENTATION DETAILS

COMPASS integrates VLMs to process multi-modal inputs and generate executable skills in two stages. Each skill follows a standardized interface:

---

**Algorithm 1:** COMPASS Agent Decision-Making Loop

---

**Initialize:**

skill_manager.bootstrap(demonstration_data)

agent_state ← environment.reset(agent_id)

local_memory ← initialize_local_memory()

global_memory ← initialize_global_memory()

previous_action_result ← None

**while** *True* **do**

    // 1.  Communication Phase

    local_observations ← agent_state.get_observations()

    communication_protocol.share_local_observations(agent_id, local_observations, global_memory)

    global_entity_info ← communication_protocol.get_global_memory.update(global_memory)

    // 2.  Perception Phase

    processed_state ← vlm_perception.process( raw_observation=agent_state, communication_data=global_entity_info, local_memory=local_memory )

    local_memory.update(processed_state)

    // 3.  Self-Reflection Phase

    **if** *previous_action_result ≠ None* **then**

        reflection_feedback ← vlm_self_reflection.reflect( previous_action_result )

    // 4.  Task Reasoning Phase

    sub_task ← vlm_task_reasoning.props_subtask( processed_state, overall_goal, reflection_feedback )

    // 5.  Skill Generation Phase

    new_skill_code ← vlm_skill_generator.generate_skill(sub_task, processed_state)

    skill_manager.add_skill(new_skill_code, sub_task)

    // 6.  Actor Phase

    relevant_skills ← skill_manager.retrieve_skills(sub_task)

    chosen_skill_code ← vlm_actor.select_skill( sub_task, relevant_skills, processed_state )

    // 7.  Execution Phase

    (next_agent_state, reward, done, info) ← environment.step( agent_id, chosen_skill_code )

    agent_state ← next_agent_state

    previous_action_result ← (chosen_skill_code, info, reward, done)

    local_memory.add_action(chosen_skill_code)

    **if** *done* **then**

        **break**

---

Listing 1: Interface of Generated skills.

```
def skill_template(obs: str):
    obs_data = parse_obs(obs)
    def score_target(unit):
        ...
        return score
    def control_logic():
        ...
        return atomic_action
```

The skills follow a structured decision-making pipeline with two core components: score_target(unit) and control_logic():

- score_target(unit): Dynamically calculates a threat/priority score by evaluating unit type, health, distance, formations, and matchups to guide optimal attack/heal targeting decisions.
- control_logic(): Dynamically coordinates unit behavior by integrating observations, target priorities, and pathfinding to execute role-optimized strategies (e.g., stalkers attack colossus at max range while moving away from zealots).

COMPASS evolves skills through iterative refinement and task-guided synthesis:

- iterative refinement: When errors occur during skill execution, VLMs analyze the error messages and attempt to fix the bugs.
- task-guided synthesis: When a new task is proposed, VLMs first determine whether a new skill needs to be generated to align with the task. If necessary, VLMs generate new score_target or control_logic components to fulfill the task requirements and integrate them with the existing code to construct a new skill.

For example, if the task is: "*Implement an aggressive advance movement pattern for the colossus unit when enemy stalkers are within sight range and allies are positioned to provide covering fire*," and the current skills in the library are not aggressive enough, the VLMs will refine the control_logic to implement a more aggressive behavior pattern.

COMPASS maintains a global entity memory, where agents act as nodes connected if within sight range. Each node stores information about visible allies and enemies. When Agent A observes entities, it propagates updates through adjacent nodes recursively, up to max_hops (default = 3). This allows agents to infer off-screen threats via ally intermediaries. For implementation details, see ./common/memory/global_memory.py.

## C ENVIRONMENT SETTINGS AND MORE RESULTS

Table 5: SMACv2 task scenarios

| TASK | SCENARIOS | CATEGORIES |
|---|---|---|
| PROTOSS | PROTOSS 5 VS 5 | SYMMETRIC |
| | PROTOSS 5 VS 6 | ASYMMETRIC |
| TERRAN | TERRAN 5 VS 5 | SYMMETRIC |
| | TERRAN 5 VS 6 | ASYMMETRIC |
| ZERG | ZERG 5 VS 5 | SYMMETRIC |
| | ZERG 5 VS 6 | ASYMMETRIC |

## D SKILL ANALYSIS

We now analyze COMPASS's capability to synthesize and execute diverse tactical behaviors. COMPASS develops four key tactical patterns: (1) An exponentially-scaled focus fire implementation

Table 6: We report quantitative results on SMACv2 under sparse reward settings, excluding COM-PASS due to its inherent insensitivity to reward sparsity.

| | $\text{QMIX}_s$ | $\text{MAPPO}_s$ | $\text{HAPPO}_s$ | $\text{HASAC}_s$ |
|---|---|---|---|---|
| | PROTOSS | | | |
| 5V5 | 0 | 0 | 0 | 0 |
| 5V6 | 0 | 0 | 0 | 0 |
| | TERRAN | | | |
| 5V5 | 0 | 0 | 0 | 0 |
| 5V6 | 0 | 0 | 0 | 0 |
| | ZERG | | | |
| 5V5 | $0.02_{0.01}$ | 0 | 0 | 0 |
| 5V6 | 0 | 0 | 0 | 0 |

that coordinates multiple units' target selection based on allied attacker density (Figure 6), (2) A position-aware kiting mechanism that maintains optimal engagement ranges while managing unit positioning relative to threats (Figure 7), (3) A formation-based isolation tactic that enables systematic target elimination through coordinated unit movements (Figure 8), and (4) An area-of-effect (AOE) tactic that maximizes splash damage through cluster density calculation (Figure 9). These synthesized skills exhibit clear strategic intent while maintaining interpretability.

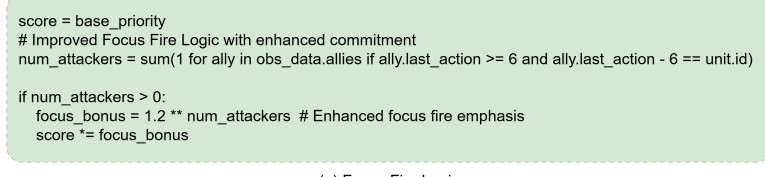

```
score = base_priority
# Improved Focus Fire Logic with enhanced commitment
num_attackers = sum(1 for ally in obs_data.allies if ally.last_action >= 6 and ally.last_action - 6 == unit.id)

if num_attackers > 0:
    focus_bonus = 1.2 ** num_attackers  # Enhanced focus fire emphasis
    score *= focus_bonus
```

(a) Focus Fire Logic

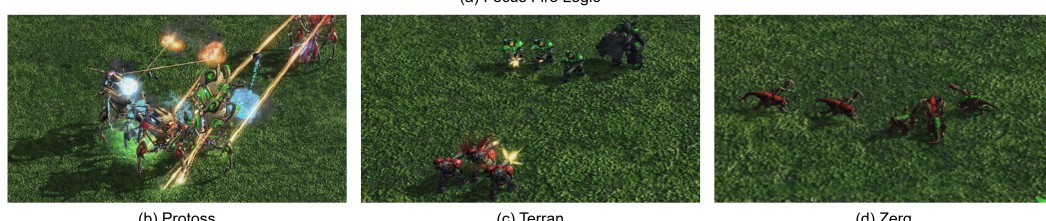

(b) Protoss          (c) Terran          (d) Zerg

Figure 6: Focus Fire Logic Implementation. (a) VLM-generated Python code snippet implementing dynamic focus fire logic. The code prioritizes enemy units based on the number of allied attackers, scaling the attack bonus exponentially. (b–d) Visualizations of focus fire execution across Protoss, Terran, and Zerg.

# E  BASELINE TRAINING RESULTS

For QMIX, we utilized Epymarl[3], and for others, we used HARL[4]. Task difficulty was set to 5v5 for SYMMETRIC tasks and 5v6 for ASYMMETRIC tasks; the 5v6 scenario introduces a significant force disadvantage (20% outnumbered vs 10% in 10v11).

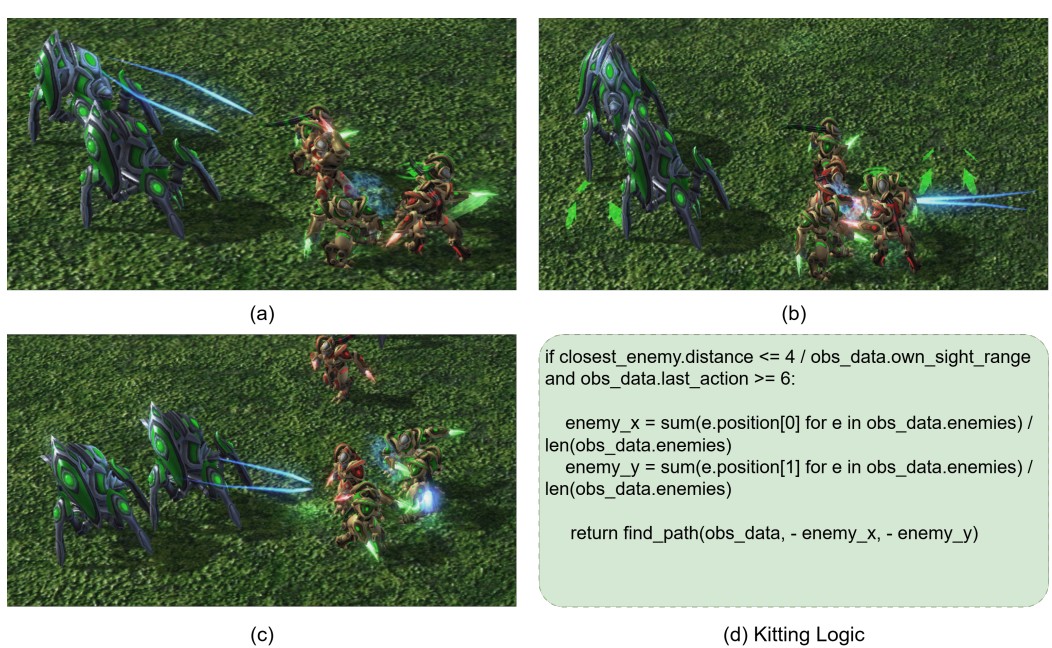

(a)             (b)

(c)           (d) Kitting Logic

```python
if closest_enemy.distance <= 4 / obs_data.own_sight_range
and obs_data.last_action >= 6:

    enemy_x = sum(e.position[0] for e in obs_data.enemies) /
len(obs_data.enemies)
    enemy_y = sum(e.position[1] for e in obs_data.enemies) /
len(obs_data.enemies)

    return find_path(obs_data, - enemy_x, - enemy_y)
```

Figure 7: Illustration and implementation of Kitting logic. (a)-(c) demonstrate progressive stages of the kitting tactic where allied units strategically maintain optimal attack range while retreating from melee enemies. (d) shows the corresponding Python code snippet generated by the VLMs.

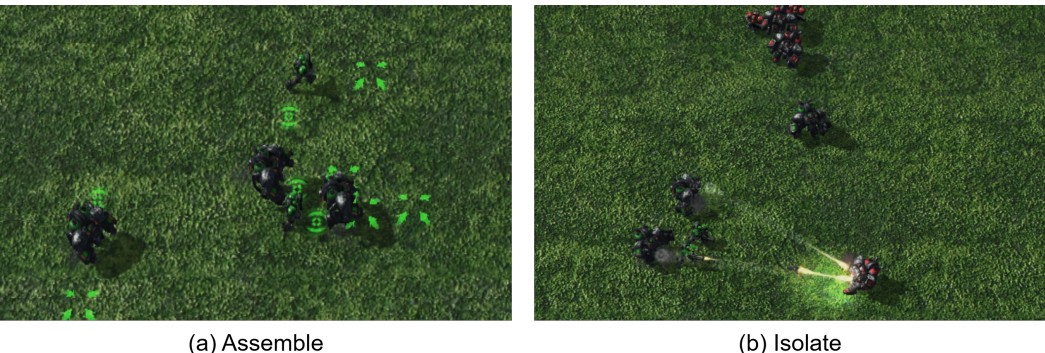

(a) Assemble           (b) Isolate

Figure 8: Illustration of Isolating logic. (a) Allied units strategically assemble into a cohesive formation. (b) The assembled units execute a rapid engagement against an isolated enemy unit, eliminating it before reinforcements can arrive, thus creating a numerical advantage.

```
enemy_clusters = {}
    for enemy in obs_data.enemies:
    nearby_enemies = []
    # Dynamic cluster radius based on unit type
    cluster_radius = 0.3 if obs_data.own_unit_type.lower() == 'baneling' else 0.2
    for other in obs_data.enemies:
    distance = ((other.position[0] - enemy.position[0])**2 + (other.position[1] - enemy.position[1])**2)**0.5
    if distance <= cluster_radius:
        nearby_enemies.append(other)
    enemy_clusters[enemy.id] = len(nearby_enemies)
```

(a) Baneling Cluster

(b)                                                    (c)

Figure 9: Demonstration of area-of-effect (AOE) optimization for Baneling units in SMACv2. (a) The VLM-generated Python code calculates optimal detonation positions by analyzing enemy cluster density and positions. (b-c) Visual sequence showing Baneling execution, where the unit identifies a dense cluster of enemy units and detonates for maximum AOE damage.

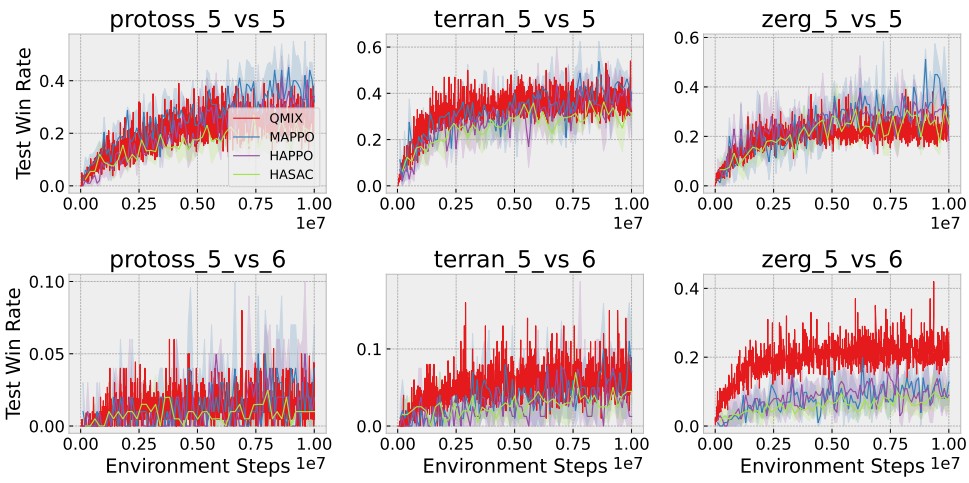

Figure 10: Baseline training results on SMACv2.

| Method | Type | Protoss 5v5 | Protoss 5v6 | Terran 5v5 | Terran 5v6 | Zerg 5v5 | Zerg 5v6 |
|---|---|---|---|---|---|---|---|
| **COMPASS** | **VLMs** | **0.57 ± 0.08** | **0.08 ± 0.04** | **0.39 ± 0.01** | **0.10 ± 0.03** | 0.16 ± 0.07 | 0.03 ± 0.01 |
| MAT | Online | 0.39 ± 0.03 | 0.04 ± 0.04 | 0.36 ± 0.11 | 0.05 ± 0.01 | **0.32 ± 0.06** | 0.11 ± 0.08 |
| CommFormer | Online | 0.39 ± 0.16 | 0.02 ± 0.01 | 0.30 ± 0.09 | 0.03 ± 0.01 | **0.39 ± 0.10** | **0.16 ± 0.01** |
| Oryx | Offline | N/A* | N/A* | 0.18 ± 0.04 | N/A* | 0.10 ± 0.06 | N/A* |

Table 7: Comparison of methods across SMACv2 scenarios. N/A indicates no datasets in these settings.

## F  MORE RESULTS

## G  TOKEN USAGE ANALYSIS

We acknowledge the computational cost. We provide a breakdown of token usage per step per agent (call frequency: 20).

- **Average Episode Total Tokens:** 80,000
- **Average Tokens per Step:** 1,333 (Perception: 2k, Actor: 2k, Self-Reflection: 2k, Task Reasoning: 10k, Skill Synthesis: 10k)

The bottlenecks are **Task Reasoning** and **Skill Synthesis**, which require historical context and code history. Skill synthesis often requires multiple rounds of self-correction to generate executable code. Future work will focus on reducing this cost by exploiting structural symmetries among agents. We may also reduce token usage by appropriately lowering the frequency of queries.

---

[3]https://github.com/uoe-agents/epymarl
[4]https://github.com/PKU-MARL/HARL

## H    PROMPTS USED IN COMPASS

---

**Prompt for Perception**

You are an AI assistant helping with academic research in the StarCraft II's SMAC (StarCraft Multi-Agent Challenge) environment, controlling a <unit_type> unit with ID <unit_id> in micromanagement scenarios <scenario_name> to help your team defeat the enemy forces. You operate under decentralized execution with partial observability, making decisions based only on local observations within your unit's field of view. Your advanced capabilities enable you to process and interpret gameplay screenshots and other relevant information.
I will give you the following information:
<few_shots>
Reasoning for the last episode:
<last_episode_reasoning>
Strategic situation analysis:
<info_summary>
Below is the current in-game screenshot and its description:
<image_introduction>
Minimap information:
<ego_minimap>
Current task:
<task_description>
Tactics recommendation:
<web_search>
Based on the above information, you should first analyze the current game situation by integrating the information from the in-game screenshot, its description, and other provided information.
Game situation:
You should think step by step and provide detailed reasoning to determine the current state of the game.
You need to answer the following questions step by step:
1. What is your unit_id, unit type?
2. What map borders are you near? Check which cardinal directions (N/S/E/W) have unavailable movement actions.
3. What is the current health status of your unit? What is the current shield status of your unit?
4. Are there any enemy units visible, either in observation or minimap?
5. Are there any ally units visible, either observation or minimap?
6. Are you positioned at the optimal attack range from enemies, or do you need to reposition based on the enemies' locations and directions?
Region of interest:
What unit or location should be interacted with to complete the task based on the current screenshot and the current task? You should obey the following rules:
1. If your chosen region of interest is a unit, format the output as "[Enemy/Ally] #[target_id]" (e.g., "Enemy #0" for enemy unit with ID 0, "Ally #1" for ally unit with ID 1)
2. If your chosen region of interest is location, format the output as "Location: [direction]" where direction must be one of: "North", "Northeast", "East", "Southeast", "South", "Southwest", "West", "Northwest", "Center" (e.g., "Location: Northeast")
3. If there are units visible, prioritize using unit as region of interest.
4. If the target_id is required, you MUST only use enemy/ally's unit_ids that are currently visible in your shooting range.
5. If your chosen region of interest is location, you MUST verify its availability.
6. If shared minimap information reveals enemies outside your sight range, prioritize moving to those locations unless there are enemies within your current vision range.
7. Your chosen region of interest should align with the current task description and ally's intentions.
8. Your chosen region of interest should enable you to quickly engage in combat or efficiently achieve the task in cooperation with allies?
Reasoning of region of interest:
Why was this region of interest chosen?
You should only respond in the format described below with a line break after each section colon (##Section##:) and NOT output comments or other information:
##Game_situation##:
1. ...
##Region_of_interest##:
region of interest

---

##Reasoning_of_region_of_interest##:
1. ...

---

**Prompt for Task Reasoning**

You are an AI assistant helping with academic research in the StarCraft II's SMAC (StarCraft Multi-Agent Challenge) environment, controlling a <unit_type> unit with ID $<unit_id>$ in micromanagement scenarios <scenario_name> to help your team defeat the enemy forces. You operate under decentralized execution with partial observability, making decisions based only on local observations within your unit's field of view. You will be sequentially given <event_count> screenshots and corresponding descriptions of recent events. You will also be given a summary of the history that happened before the last screenshot. By analyzing these inputs, you gain a comprehensive understanding of the current context and situation within the game. You should assist in summarizing the next immediate task to do in SMACv2. Your ultimate goal is to help your team defeat the enemy forces as quickly as possible.
I will give you the following information:
Reasoning for the last episode:
<last_episode_reasoning>
Cumulative reward for the executing skill:
<cumulative_reward>
Current task:
<task_description>
Ally's tasks:
<ally_task>
Minimap information:
<ego_minimap>
Current game situation:
<game_situation>
Tactics recommendation:
<web_search>
The following are successive screenshots:
<image_introduction>
Skill set in Python format to select the next skill:
<skill_library>
Current executing skill:
<previous_action>
Implementation of the skill:
<action_code>
Reasoning for the skill:
<previous_reasoning>
Self-reflection for the last executed skill:
<previous_self_reflection_reasoning>
Task_guidance:
Based on the comprehensive game state analysis and team context, decompose the primary objective of "defeat all enemy units" into ONE specific tactical sub-task that enhances either target prioritization (score_target) or behavior control (control_logic). This sub-task should be concrete, implementable, and aligned with team coordination. Consider the following in your task decomposition:
1. Final Objective: Defeat enemy forces while preserving allies
2. Team Context:
- Your unit's current assigned task - Ally units' assigned tasks - Progress made on previous tasks 3. Tactical Layer:
- Enemy unit compositions and strategies - Team formation and positioning The task should follow one of these formats:
For target prioritization (score_target):
"Adjust [scoring weight/multiplier/threshold] to [specific combat calculation] based on [unit composition + battle state] where [precise condition]"
For behavior control (control_logic):
"Implement [unit movement pattern/formation/targeting] when [combat state + ally positions] satisfy [precise conditions]" Task Requirements:
Specificity: Must define exact behavior modification Measurability: Must have clear success criteria Actionability: Must be achievable using available atomic actions Coordination: Must support team tactical objectives Adaptability: Must respond to changing battle conditions If current task implementation remains unsuccessful, output 'null'.

Reasoning_of_task:
Why was this new task chosen, or why is there no need to propose a new task?
Skill_guidance:
Based on the current executing skill and the proposed next task, evaluate if there is alignment between them. Output True if the current skill effectively supports the task requirements, or False if a new skill is needed.
Reasoning_of_skill:
Why was this decision chosen?
You should only respond in the format described below with a line break after each section colon (##Section##:) and NOT output comments or other information:
##Task_guidance##:
[task guidance]
##Skill_guidance##:
[True or False]

---

## Prompt for Skill Generation

You are an AI assistant helping with academic research in the StarCraft II's SMAC (StarCraft Multi-Agent Challenge) environment, controlling a <unit_type> unit with ID $<unit_id>$ in micromanagement scenarios <scenario_name> to help your team defeat the enemy forces. You operate under decentralized execution with partial observability, making decisions based only on local observations within your unit's field of view. Your task is to enhance combat effectiveness:
Reasoning for the last episode:
<last_episode_reasoning>
Cumulative reward for the executing skill:
<cumulative_reward>
Current task:
<task_description>
Ally's tasks:
<ally_task>
Minimap information:
<ego_minimap>
Current game situation:
<game_situation>
<image_introduction>
Skill set in Python format to select the next skill:
<skill_library>
Current executing skill:
<previous_action>
Implementation of the skill:
<action_code>
Reasoning for the skill:
<previous_reasoning>
Self-reflection for the last executed skill:
<previous_self_reflection_reasoning>
Combat Analysis Task:
1. Analyze the provided script's effectiveness
2. Analyze the score_target(unit) function's effectiveness and weaknesses.
3. Analyze the control_logic() function's effectiveness and weaknesses.
4. Based on the current executing skill, the existing skills in skill library, and current task, evaluate if there is alignment between them.
5. If a new skill is needed, design tactical improvements while maintaining code structure.
6. If the current skill or there is any skill in skill library effectively supports the task requirements, output 'null' to avoid unnecessary token consumption.
Identify critical function for improvement (choose ONE Prioritize score_target(unit)):
1. score_target(unit): Target priority and scoring system. (Preferred)
2. control_logic(): Unit movement and attack decision making.
Skill_generation:
If there is no enemies, only output 'null'.
If the current skill or there is any skill in skill library effectively supports the task requirements, only output 'null'.
Otherwise:

The content of the improved code should obey the following code rules:
1. Output Format: Only provide the complete improved function (score_target(unit) (Preferred) OR control_logic()).
2. If the improved function is score_target(unit), there is exactly one parameter named "unit".
3. If the improved function is control_logic(), it should take no parameters.
4. The code should be surrounded in the '"'python' and '"' structure.

You should only respond in the format described below with a line break after each section colon (##Section##:) and NOT output comments or other information:

##Skill_generation##:
'"python
def [function_name]([parameters]):
[improved implementation] '"

---

### Prompt for Actor

You are an AI assistant helping with academic research in the StarCraft II's SMAC (StarCraft Multi-Agent Challenge) environment, controlling a <unit_type> unit with ID <$unit_id$> in microman-agement scenarios <scenario_name> to help your team defeat the enemy forces. You operate under decentralized execution with partial observability, making decisions based only on local observations within your unit's field of view. Utilizing this insight, you are tasked with identifying the most suitable skill to take next, given the current task. You control the game unit and can execute skills from the available skill set. Upon evaluating the provided information, your role is to articulate the precise skill you would deploy, considering the game's present circumstances, and specify any necessary parameters for implementing that skill:
<last_episode_reasoning>
Cumulative reward for the executing skill:
<cumulative_reward>
Current task:
<task_description>
Ally's tasks:
<ally_task>
Minimap information:
<ego_minimap>
Current game situation:
<game_situation>
<image_introduction>
Skill set in Python format to select the next skill:
<skill_library>
Current executing skill:
<previous_action>
Implementation of the skill:
<action_code>
Reasoning for the skill:
<previous_reasoning>
Self-reflection for the last executed skill:
<previous_self_reflection_reasoning>
Skills:
The best skill to execute next to progress in achieving the goal. Pay attention to the names of the available skills and to the previous skills already executed, if any. You should also pay more attention to the following skill rules:
1. ONLY choose skill in the provided skill set.
2. Output skills in Python code format with required keyword parameters.
3. The ONLY required keyword parameter is "obs: str" - you MUST include this parameter as "obs='current'" in every skill. The actual observation will be automatically injected at runtime.
4. If there is summarization of history, consider this information when selecting the skill.
5. If the error report indicates that the last skill was unavailable, you MUST select a different skill.
6. Consider coordination with other units and choose skills that enhance team performance and cooperation.
7. Avoid repeating the same skill as the last executed skill unless there is a compelling strategic reason.
You should only respond in the format described below with a line break after each section colon

```python
def race_melee_ranged_medivac_navi_A_star_score_type_default_center(obs:
    ↪ str):
    """

    ↪ Zealot/Zergling/Baneling/Colossus/Stalker/Hydralisk/Marauder/Marine/Medivac
    ↪ Controls Logic:
        Medivac:
            - Heals allies below 100% HP
            - Maintains 0.75 sight range from enemies
            - Centers between allies when no healing targets

        Melee (Zealot/Zergling/Baneling):
            - Attacks highest threat target within 0.7 sight range
            - Pursues targets using A* pathfinding
            - Groups with allies at >0.7 distance threshold

        Ranged (Colossus/Stalker/Hydralisk/Marauder/Marine):
            - Kites melee units at max_range - 0.05
            - Focus fires targets shared with 2+ allies
            - Maintains position behind melee allies

        Key Implementation:
        - Pathfinding: A* pathfinding in 32x32 grid with unit collision
    ↪ radius
        - Target scoring: [0-10] based on type (colossus 9 > baneling 8
    ↪ > zealot 7 > stalker 6 > hydralisk 5 > marauder 4 > marine 3 >
    ↪ zergling 2 > medivac 1)/health (0-0.6)/distance (0-0.3)/last
    ↪ attacked (0.1)
        - Default action: Move to center of map, parse region of
    ↪ interest, random choice

    Args:
        obs (str): Observation string containing game state
    """
    import math
    # Parse observation
    obs_data = parse_obs(obs)
    # Get set of available actions
    valid_actions = obs_data.available_actions

    if 0 in valid_actions:
        return 0

    def score_target(unit):
        """Enhanced target scoring with improved kiting and formation
    ↪ control"""
        if unit.health <= 0:
            return -1

        score = 0

        # Refined unit type priorities with enhanced threat scaling
        unit_priorities = {
            'colossus': 35.0,  # Further increased priority
            'stalker': 30.0,   # Enhanced anti-armor focus
            'zealot': 45.0,    # Higher melee threat recognition

```

```python
49              'marine': 45.0,     # Balanced damage dealer priority
50              'marauder': 35.0,   # Anti-armor specialist
51              'medivac': 30.0,    # Support unit priority
52
53              'hydralisk': 30.0,  # High priority for their sustained DPS
54              'zergling': 35.0,   # Medium priority as swarm units
55              'baneling': 45.0,   # Critical priority due to splash damage
56          }
57
58          # Dynamic matchup priorities with improved counter weighting
59          unit_counters = {
60              'colossus': {'colossus': 1.2, 'stalker': 1.0, 'zealot': 1.5},
61              'stalker': {'colossus': 1.2, 'stalker': 1.0, 'zealot': 1.5},
62              'zealot': {'colossus': 1.2, 'stalker': 1.0, 'zealot': 1.5},
63
64              'marine': {'marine': 1.5, 'medivac': 1.0, 'marauder': 1.2},
65              'marauder': {'marine': 1.5, 'medivac': 1.0, 'marauder': 1.2},
66              'medivac': {'marine': 1.5, 'medivac': 1.0, 'marauder': 1.2},
67
68              'hydralisk': {'hydralisk': 1.0, 'zergling': 1.2, 'baneling':
    ↪ 1.5},
69              'zergling': {'hydralisk': 1.2, 'zergling': 1.5, 'baneling':
    ↪ 1.0},
70              'baneling': {'hydralisk': 1.2, 'zergling': 1.5, 'baneling':
    ↪ 1.0},
71
72          }
73
74          base_priority = unit_priorities.get(unit.unit_type.lower(), 5.0)
75          is_ranged = unit.unit_type.lower() not in ['zealot', 'zergling',
    ↪ 'baneling']
76          own_is_ranged = obs_data.own_unit_type.lower() not in ['zealot',
    ↪ 'zergling', 'baneling']
77
78          # Enhanced threat assessment with improved melee handling
79          matchup_mult = unit_counters.get(obs_data.own_unit_type.lower(),
    ↪ {}).get(unit.unit_type.lower(), 1.0)
80          base_priority *= matchup_mult
81
82          if hasattr(unit, 'can_attack'):  # Enemy unit
83              score = base_priority
84
85              distance_factor = max((1 - unit.distance) + 1, 0.5)
86              score *= distance_factor
87
88              if not own_is_ranged:
89                  range_ally = [ally for ally in obs_data.allies if
    ↪ ally.unit_type.lower() not in ['zealot', 'zergling', 'baneling']]
90                  if range_ally:
91                      ally_x = sum(ally.position[0] for ally in
    ↪ range_ally) / len(range_ally)
92                      ally_y = sum(ally.position[1] for ally in
    ↪ range_ally) / len(range_ally)
93                      ally_distance = ((ally_x - unit.position[0])**2 +
    ↪ (ally_y - unit.position[1])**2)**0.5
94                      distance_factor = max((1 - ally_distance) + 1, 0.5)
95                      score *= distance_factor
96
97              # Enhanced Position Analysis with improved spacing
98              position_x, position_y = unit.position
99
100             def calculate_combat_power(units, radius=0.5):  # Further
    ↪ reduced for tighter control
101                 total_power = 0
102                 ranged_count = 0
```

```
103                     melee_count = 0
104                     unit_positions = []
105
106                     for u in units:
107                         dist = ((u.position[0] - position_x)**2 +
108                             (u.position[1] - position_y)**2)**0.5
109                         unit_positions.append(u.position)
110
111                         if dist <= radius:
112                             base_power =
    ↪ unit_priorities.get(u.unit_type.lower(), 5.0)
113
114                             # Unit type specific power calculation
115                             if u.unit_type.lower() in ['zealot', 'zergling',
    ↪ 'baneling']:
116                                 melee_count += 1
117                                 if melee_count >= 2:
118                                     base_power *= 1.4
119                             else:
120                                 ranged_count += 1
121                                 base_power *= 1.3
122
123                             # Health-based power scaling
124                             health_factor = 1.5 if u.health > 0.7 else 1.0
    ↪ if u.health > 0.4 else 0.6
125                             position_factor = 1.3 - (dist/radius)
126
127                             total_power += base_power * health_factor *
    ↪ position_factor
128
129                     # Enhanced formation cohesion calculation
130                     cohesion = 0
131                     if len(unit_positions) > 2:
132                         center_x = sum(p[0] for p in unit_positions) /
    ↪ len(unit_positions)
133                         center_y = sum(p[1] for p in unit_positions) /
    ↪ len(unit_positions)
134                         avg_dist = sum(((p[0] - center_x)**2 + (p[1] -
    ↪ center_y)**2)**0.5
135                                     for p in unit_positions) /
    ↪ len(unit_positions)
136                         max_desired_dist = 0.3  # Tighter formation control
137                         cohesion = 2.0 / (1.0 + (avg_dist /
    ↪ max_desired_dist))
138
139                     return total_power * (1 + cohesion), melee_count,
    ↪ ranged_count
140
141             ally_power, ally_swarms, ally_ranged =
    ↪ calculate_combat_power(obs_data.allies)
142             enemy_power, enemy_swarms, enemy_ranged =
    ↪ calculate_combat_power(obs_data.enemies)
143
144             # Improved Focus Fire Logic with enhanced commitment
145             num_attackers = sum(1 for ally in obs_data.allies
146                             if ally.last_action >= 6 and
    ↪ ally.last_action - 6 == unit.id)
147
148             if unit.id == obs_data.last_action - 6:
149                 persistence_bonus = 2.0  # Stronger target commitment
150                 score *= persistence_bonus
151
152             if num_attackers > 0:
153                 focus_bonus = 1.2 ** num_attackers  # Enhanced focus
    ↪ fire emphasis
```

```
154                     # if num_attackers too high, discourage prevent
    ↪ overcommitment
155                     if num_attackers >= 3 and unit.id !=
    ↪ obs_data.last_action - 6 and obs_data.own_unit_type.lower() in
    ↪ ['zergling', 'baneling']:
156                         focus_bonus = 0.5
157                     score *= focus_bonus
158
159                 # Improved Combat Advantage Factor
160                 advantage_factor = 1.0
161                 if ally_power > enemy_power * 1.3:
162                     advantage_factor = 1.2  # More aggressive advantage
    ↪ pursuit
163                     if ally_swarms >= 3:
164                         advantage_factor *= 1.2
165
166                 # prioritize isolated enemies
167                 if (enemy_swarms + enemy_ranged) == 1:
168                     advantage_factor *= 2.0
169                 elif (ally_swarms + ally_ranged) > (enemy_swarms +
    ↪ enemy_ranged):
170                     advantage_factor *= 1.2
171
172                 score *= advantage_factor
173
174                 # Health factor
175                 health_factor = (1 - unit.health) + 1
176                 score *= health_factor
177
178             else:  # Ally unit
179                 score = base_priority
180
181                 # Improved Support Priority
182                 health_factor = (1 - unit.health) + 1
183                 score *= health_factor
184
185                 distance_factor = max((1 - unit.distance) + 1, 0.5)
186                 score *= distance_factor
187
188             return score
189
190         def control_logic():
191             # Medivac units control logic
192             if obs_data.own_unit_type.lower() == 'medivac':
193                 attack_actions = [a for a in valid_actions if a >= 6]
194                 # If there are allies
195                 if obs_data.allies:
196                     lowest_health_ally = min(obs_data.allies, key=lambda x:
    ↪ x.health)
197                     # If there are both allies and enemies
198                     if obs_data.enemies:
199                         enemy_in_range = [enemy for enemy in
    ↪ obs_data.enemies if enemy.distance < 1]
200                         # Check if any melee ally, if so and last action is
    ↪ not attack, move to center of melee allies
201                         melee_ally = [ally for ally in obs_data.allies if
    ↪ ally.unit_type.lower() in ['zealot', 'zergling', 'baneling']]
202                         if melee_ally:
203                             # Move to center of melee allies
204                             ally_x = sum(ally.position[0] for ally in
    ↪ melee_ally) / len(melee_ally)
205                             ally_y = sum(ally.position[1] for ally in
    ↪ melee_ally) / len(melee_ally)
206                         else:
```

```
207                         ally_x = sum(ally.position[0] for ally in
    ↪ obs_data.allies) / len(obs_data.allies)
208                         ally_y = sum(ally.position[1] for ally in
    ↪ obs_data.allies) / len(obs_data.allies)
209                     # Calculate retreat position
210                     if len(enemy_in_range) > 0:
211                         enemy_center = (sum(e.position[0] for e in
    ↪ enemy_in_range) / len(enemy_in_range),
212                                         sum(e.position[1] for e in
    ↪ enemy_in_range) / len(enemy_in_range))
213                     else:
214                         enemy_center = (sum(e.position[0] for e in
    ↪ obs_data.enemies) / len(obs_data.enemies),
215                                         sum(e.position[1] for e in
    ↪ obs_data.enemies) / len(obs_data.enemies))
216
217                     dx = ally_x - enemy_center[0]
218                     dy = ally_y - enemy_center[1]
219
220                     distance = (dx ** 2 + dy ** 2) ** 0.5
221
222                     safe_x = ally_x + (dx / abs(dx)) *
    ↪ 2/obs_data.own_sight_range if dx != 0 else ally_x
223                     safe_y = ally_y + (dy / abs(dy)) *
    ↪ 2/obs_data.own_sight_range if dy != 0 else ally_y
224
225                     distance = (safe_x ** 2 + safe_y ** 2) ** 0.5
226                     criterion = 5/obs_data.own_sight_range
227                     if obs_data.last_action >= 6 or len(attack_actions)
    ↪ == 0:
228                         target_angle = math.atan2(enemy_center[1],
    ↪ enemy_center[0])
229                         safe_angle = math.atan2(ally_y, ally_x)
230                         angle_diff = abs(target_angle - safe_angle)
231                         if distance > criterion and (math.pi/9 <
    ↪ angle_diff < 17*math.pi/9):
232                             path_action = find_path(obs_data, safe_x,
    ↪ safe_y)
233                             if path_action:
234                                 return path_action
235                 target_scores = {ally.id: score_target(ally) for ally in
    ↪ obs_data.allies}
236                 # Check if there are same max score targets
237                 max_score = max(target_scores.values())
238                 max_score_target_ids = [target_id for target_id, score
    ↪ in target_scores.items() if score == max_score]
239                 max_score_targets = [ally for ally in obs_data.allies if
    ↪ ally.id in max_score_target_ids]
240                 closest_ally = min(obs_data.allies, key=lambda x:
    ↪ x.distance)
241                 if len(max_score_targets) > 1:
242                     # Chose the closest target
243                     best_target = min(max_score_targets, key=lambda x:
    ↪ x.distance)
244                 else:
245                     best_target = max_score_targets[0]
246                 if (best_target.id + 6) in valid_actions and 0 <
    ↪ best_target.health < 0.9:
247                     return heal(best_target.id)
248                 elif (closest_ally.id + 6) in valid_actions and 0 <
    ↪ closest_ally.health < 0.9:
249                     return heal(closest_ally.id)
250                 elif (lowest_health_ally.id + 6) in valid_actions and 0
    ↪ < lowest_health_ally.health < 0.9:
251                     return heal(lowest_health_ally.id)
```

```
252                     else:
253                         # Move to the target
254                         dx = best_target.position[0]
255                         dy = best_target.position[1]
256                         path_action = find_path(obs_data, dx, dy,
      ↪ target_type=best_target.unit_type.lower())
257                         if path_action:
258                             return path_action
259             # If there are no allies
260             else:
261                 # If there are only enemies
262                 if obs_data.enemies:
263                     enemy_x = sum(e.position[0] for e in
      ↪ obs_data.enemies) / len(obs_data.enemies)
264                     enemy_y = sum(e.position[1] for e in
      ↪ obs_data.enemies) / len(obs_data.enemies)
265                     target_x = - enemy_x
266                     target_y = - enemy_y
267
268                     g_x = target_x * obs_data.own_sight_range +
      ↪ (obs_data.own_position[0] * 32)
269                     g_y = target_y * obs_data.own_sight_range +
      ↪ (obs_data.own_position[1] * 32)
270                     if not (0 <= g_x <= 32 and 0 <= g_y <= 32):
271                         target_x = (0.5 - obs_data.own_position[0]) * 32
      ↪ / obs_data.own_sight_range
272                         target_y = (0.5 - obs_data.own_position[1]) * 32
      ↪ / obs_data.own_sight_range
273
274                     path_action = find_path(obs_data, target_x, target_y)
275                     if path_action:
276                         return path_action
277         # Melee units control logic
278         elif obs_data.own_unit_type.lower() in ['zealot', 'zergling',
      ↪ 'baneling']:
279             # If there are enemies
280             if obs_data.enemies:
281                 enemy_in_range = [enemy for enemy in obs_data.enemies if
      ↪ enemy.distance < 1]
282                 attack_actions = [a for a in valid_actions if a >= 6]
283
284                 if len(enemy_in_range) > 0:
285                     enemy_center = (sum(e.position[0] for e in
      ↪ enemy_in_range) / len(enemy_in_range),
286                                     sum(e.position[1] for e in
      ↪ enemy_in_range) / len(enemy_in_range))
287                 else:
288                     enemy_center = (sum(e.position[0] for e in
      ↪ obs_data.enemies) / len(obs_data.enemies),
289                                     sum(e.position[1] for e in
      ↪ obs_data.enemies) / len(obs_data.enemies))
290                 if obs_data.allies:
291                     # Check if any melee ally, if so and last action is
      ↪ not attack, move to center of melee allies
292                     melee_ally = [ally for ally in obs_data.allies if
      ↪ ally.unit_type.lower() in ['zealot', 'zergling', 'baneling']]
293                     if melee_ally:
294                         # Move to center of melee allies
295                         ally_x = sum(ally.position[0] for ally in
      ↪ melee_ally) / len(melee_ally) / 2
296                         ally_y = sum(ally.position[1] for ally in
      ↪ melee_ally) / len(melee_ally) / 2
297
298                         safe_x = ally_x
299                         safe_y = ally_y
```

```
300                     distance = (safe_x ** 2 + safe_y ** 2) ** 0.5
301                     criterion = 2/obs_data.own_sight_range
302                     if len(attack_actions) == 0 or distance > 0.5:
303                         target_angle = math.atan2(enemy_center[1],
    ↪ enemy_center[0])
304
305                         safe_angle = math.atan2(ally_y, ally_x)
306                         angle_diff = abs(target_angle - safe_angle)
307                         if distance > criterion and (math.pi/9 <
    ↪ angle_diff < 17*math.pi/9 or distance > 0.5):
308                             path_action = find_path(obs_data,
    ↪ safe_x, safe_y)
309                             if path_action:
310                                 return path_action
311
312             # Enhanced cluster detection with dynamic radius
313             enemy_clusters = {}
314             cluster_centers = {}
315             for enemy in obs_data.enemies:
316                 nearby_enemies = []
317                 center_x, center_y = enemy.position[0],
    ↪ enemy.position[1]
318
319                 # Dynamic cluster radius based on unit type
320                 cluster_radius = 0.3 if
    ↪ obs_data.own_unit_type.lower() == 'baneling' else 0.2
321
322                 for other in obs_data.enemies:
323                     distance = ((other.position[0] -
    ↪ enemy.position[0])**2 +
324                                 (other.position[1] -
    ↪ enemy.position[1])**2)**0.5
325                     if distance <= cluster_radius:
326                         nearby_enemies.append(other)
327                         center_x += other.position[0]
328                         center_y += other.position[1]
329
330                 if nearby_enemies:
331                     center_x /= len(nearby_enemies)
332                     center_y /= len(nearby_enemies)
333
334                 enemy_clusters[enemy.id] = len(nearby_enemies)
335                 cluster_centers[enemy.id] = (center_x, center_y)
336
337             # Enhanced target scoring with tactical considerations
338             target_scores = {}
339             for enemy in obs_data.enemies:
340                 base_score = score_target(enemy)
341
342                 # Enhanced cluster bonus for splash damage
343                 if obs_data.own_unit_type.lower() == 'baneling':
344                     cluster_bonus = 1.5 ** enemy_clusters[enemy.id]
345                 else:
346                     cluster_bonus = 1.2 ** enemy_clusters[enemy.id]
347                 # Calculate final score with all factors
348                 target_scores[enemy.id] = (base_score +
    ↪ cluster_bonus)
349
350             # Check if there are same max score targets
351             max_score = max(target_scores.values())
352             max_score_targets = [enemy for enemy in obs_data.enemies
353                     if target_scores[enemy.id] >= max_score]  #
    ↪ Allow for close scores
354             if len(max_score_targets) > 1:
```

```
355                         # Choose target balancing distance and cluster
    ↪ potential
356                         best_target = min(max_score_targets,
357                                 key=lambda x: x.distance)
358                     else:
359                         best_target = max_score_targets[0]
360
361                     if best_target.can_attack:
362                         return attack(best_target.id)
363                     else:
364                         # Move to the target
365                         dx = best_target.position[0]
366                         dy = best_target.position[1]
367                         path_action = find_path(obs_data, dx, dy,
    ↪ target_type=best_target.unit_type.lower())
368                         if path_action:
369                             return path_action
370                         elif attack_actions:
371                             attackable_enemies = [enemy for enemy in
    ↪ obs_data.enemies if enemy.can_attack]
372                             if obs_data.last_action in attack_actions:
373                                 return obs_data.last_action
374                             if attackable_enemies:
375                                 return attack(min(attackable_enemies,
    ↪ key=lambda e: e.distance).id)
376                             return random.choice(attack_actions)
377
378             # If there are no enemies
379             else:
380                 # If there are only allies
381                 if obs_data.allies:
382                     # Improved melee group formation
383                     melee_allies = [ally for ally in obs_data.allies
384                                 if ally.unit_type.lower() in ['zealot',
    ↪ 'zergling', 'baneling']]
385                     if melee_allies:
386                         spacing = 0.1 if obs_data.own_unit_type.lower()
    ↪ == 'baneling' else 0.05
387                         # Dynamic group positioning
388                         center_x = sum(ally.position[0] for ally in
    ↪ melee_allies) / len(melee_allies)
389                         center_y = sum(ally.position[1] for ally in
    ↪ melee_allies) / len(melee_allies)
390
391                         # Calculate spread from center
392                         max_spread = max(((ally.position[0] -
    ↪ center_x)**2 +
393                                         (ally.position[1] -
    ↪ center_y)**2)**0.5
394                                         for ally in melee_allies)
395
396                         own_distance = ((center_x)**2 +
    ↪ (center_y)**2)**0.5
397
398                         if own_distance > spacing or max_spread > 0.1:
399                             # Move toward center while maintaining
    ↪ minimum spacing
400                             adjusted_x = center_x * 0.85  # Slight
    ↪ offset to prevent overcrowding
401                             adjusted_y = center_y * 0.85
402                             path_action = find_path(obs_data,
    ↪ adjusted_x, adjusted_y)
403                             if path_action:
404                                 return path_action
405                         else:
```

```
406                             ally_x = sum(ally.position[0] for ally in
    ↪ obs_data.allies) / len(obs_data.allies)
407                             ally_y = sum(ally.position[1] for ally in
    ↪ obs_data.allies) / len(obs_data.allies)
408                             distance = (ally_x ** 2 + ally_y ** 2) ** 0.5
409                             if distance > 0.05:
410                                 dx = ally_x
411                                 dy = ally_y
412                                 path_action = find_path(obs_data, dx, dy)
413                                 if path_action:
414                                     return path_action
415           # Ranged units control logic
416           else:
417               attack_actions = [a for a in valid_actions if a >= 6]
418               # If there are enemies
419               if obs_data.enemies:
420                   # If there are both allies and enemies
421                   # Calculate retreat position
422                   enemy_in_range = [enemy for enemy in obs_data.enemies if
    ↪ enemy.distance < 1]
423                   if len(enemy_in_range) > 0:
424                       enemy_center = (sum(e.position[0] for e in
    ↪ enemy_in_range) / len(enemy_in_range),
425                                       sum(e.position[1] for e in
    ↪ enemy_in_range) / len(enemy_in_range))
426                   else:
427                       enemy_center = (sum(e.position[0] for e in
    ↪ obs_data.enemies) / len(obs_data.enemies),
428                                       sum(e.position[1] for e in
    ↪ obs_data.enemies) / len(obs_data.enemies))
429                   if obs_data.allies:
430                       # Check if any melee ally, if so and last action is
    ↪ not attack, move to center of melee allies
431                       melee_ally = [ally for ally in obs_data.allies if
    ↪ ally.unit_type.lower() in ['zealot', 'zergling', 'baneling']]
432                       melee_enemy = [enemy for enemy in enemy_in_range if
    ↪ enemy.unit_type.lower() in ['zealot', 'zergling', 'baneling']]
433                       if melee_ally:
434                           # Move to center of melee allies
435                           ally_x = sum(ally.position[0] for ally in
    ↪ melee_ally) / len(melee_ally)
436                           ally_y = sum(ally.position[1] for ally in
    ↪ melee_ally) / len(melee_ally)
437
438                       else:
439                           ally_x = sum(ally.position[0] for ally in
    ↪ obs_data.allies) / len(obs_data.allies) / 2
440                           ally_y = sum(ally.position[1] for ally in
    ↪ obs_data.allies) / len(obs_data.allies) / 2
441
442                       dx = ally_x - enemy_center[0]
443                       dy = ally_y - enemy_center[1]
444                       safe_x = ally_x
445                       safe_y = ally_y
446                       if melee_ally:
447                           safe_x = safe_x + (dx / abs(dx)) *
    ↪ 2/obs_data.own_sight_range if dx != 0 else safe_x
448                           safe_y = safe_y + (dy / abs(dy)) *
    ↪ 2/obs_data.own_sight_range if dy != 0 else safe_y
449                       melee_threaten = False
450                       if melee_enemy:
451                           closest_melee_enemy = min(melee_enemy,
    ↪ key=lambda x: x.distance)
452                           if closest_melee_enemy.distance <=
    ↪ 4/obs_data.own_sight_range:
```

```
                                    melee_threaten = True
                                    dx = safe_x - closest_melee_enemy.position[0]
                                    dy = safe_y - closest_melee_enemy.position[1]
                                    safe_x = safe_x + (dx / abs(dx)) *
        1/obs_data.own_sight_range if dx != 0 else safe_x
                                    safe_y = safe_y + (dy / abs(dy)) *
        1/obs_data.own_sight_range if dy != 0 else safe_y

                        distance = (safe_x ** 2 + safe_y ** 2) ** 0.5
                        criterion = 4/obs_data.own_sight_range
                        if obs_data.last_action >= 6 or len(attack_actions)
        == 0 or distance > 0.9:
                            target_angle = math.atan2(enemy_center[1],
        enemy_center[0])
                            safe_angle = math.atan2(ally_y, ally_x)
                            angle_diff = abs(target_angle - safe_angle)
                            if distance > criterion and ((math.pi/9 <
        angle_diff < 17*math.pi/9) or melee_threaten or distance > 0.9):
                                path_action = find_path(obs_data, safe_x,
        safe_y)
                                if path_action:
                                    return path_action
                        # Focus fire logic
                        # Count how many allies are attacking each enemy
                        target_counts = {}
                        for ally in obs_data.allies:
                            if ally.last_action >= 6:
                                target_id = ally.last_action - 6
                                target_counts[target_id] =
        target_counts.get(target_id, 0) + 1
                        # Distance to safe point of each enemy affect target
        choosing
                        enemy_safe_distance = {enemy.id: ((enemy.position[0]
        - safe_x) ** 2 + (enemy.position[1] - safe_y) ** 2) ** 0.5 for
        enemy in obs_data.enemies}
                        # Find best target combining focus fire and threat
        scoring
                        target_scores = {enemy.id: score_target(enemy) for
        enemy in obs_data.enemies}
                        for target_id, count in target_counts.items():
                            if target_id in target_scores:
                                target_scores[target_id] += count * 0.5
                        for target_id, scores in target_scores.items():
                            target_scores[target_id] = scores * (1 -
        enemy_safe_distance[target_id] * 0.3)
                        best_target_id = max(target_scores.items(),
        key=lambda x: x[1])[0]
                        best_target = next(enemy for enemy in
        obs_data.enemies if enemy.id == best_target_id)
                        if best_target.can_attack:
                            return attack(best_target_id)
                        else:
                            # Best target is not in shoot range, move to
        target
                            dx = best_target.position[0]
                            dy = best_target.position[1]

                            # Only move to target if its direction is not
        conflicting with the safe point
                            # Check if target direction aligns with safe
        point direction
                            target_angle = math.atan2(dy, dx)
                            safe_angle = math.atan2(ally_y, ally_x)
                            angle_diff = abs(target_angle - safe_angle)
```

```
499                              # Only move if angle difference is less than 90
    ↪ degrees
500                              if angle_diff < math.pi/9 or angle_diff >
    ↪ 17*math.pi/9 or not melee_ally:
501                                  if best_target.distance >
    ↪ obs_data.own_shoot_range / obs_data.own_sight_range:
502                                      path_action = find_path(obs_data, dx,
    ↪ dy, target_type=best_target.unit_type.lower())
503                                      if path_action:
504                                          return path_action
505                              if attack_actions:
506                                  if obs_data.last_action in attack_actions:
507                                      return obs_data.last_action
508                                  attackable_enemies = [enemy for enemy in
    ↪ obs_data.enemies if enemy.can_attack]
509                                  closest_enemy = min(
510                                      [enemy for enemy in attackable_enemies],
511                                      key=lambda enemy: enemy.distance,
512                                      default=None,
513                                  )
514                                  if closest_enemy and
    ↪ closest_enemy.can_attack:
515                                      return attack(closest_enemy.id)
516                                  return random.choice(attack_actions)
517                              else:
518                                  if distance > criterion:
519                                      path_action = find_path(obs_data,
    ↪ safe_x, safe_y)
520                                      if path_action:
521                                          return path_action
522
523                      # If there are only enemies
524                      else:
525                          # Closest enemy as target
526                          closest_enemy = min(
527                              [enemy for enemy in obs_data.enemies],
528                              key=lambda enemy: enemy.distance,
529                              default=None,
530                          )
531                          # No allies, kitting melee enemies
532                          if closest_enemy.unit_type.lower() in ['zealot',
    ↪ 'zergling', 'baneling']:
533                              if closest_enemy.distance <= 4 /
    ↪ obs_data.own_sight_range and obs_data.last_action >= 6:
534                                  enemy_x = sum(e.position[0] for e in
    ↪ obs_data.enemies) / len(obs_data.enemies)
535                                  enemy_y = sum(e.position[1] for e in
    ↪ obs_data.enemies) / len(obs_data.enemies)
536                                  target_x = - enemy_x
537                                  target_y = - enemy_y
538
539                                  g_x = target_x * obs_data.own_sight_range +
    ↪ (obs_data.own_position[0] * 32)
540                                  g_y = target_y * obs_data.own_sight_range +
    ↪ (obs_data.own_position[1] * 32)
541                                  if not (0 <= g_x <= 32 and 0 <= g_y <= 32):
542                                      target_x = (0.5 -
    ↪ obs_data.own_position[0]) * 32 / obs_data.own_sight_range
543                                      target_y = (0.5 -
    ↪ obs_data.own_position[1]) * 32 / obs_data.own_sight_range
544
545                                  path_action = find_path(obs_data, target_x,
    ↪ target_y)
546                                  if path_action:
547                                      return path_action
```

```
548                             if closest_enemy.can_attack:
549                                 return attack(closest_enemy.id)
550                         else:
551                             # No melee enemies, highest priority enemy as
    ↪ target
552                             target_scores = {enemy.id: score_target(enemy)
    ↪ for enemy in obs_data.enemies}
553                             # Check if there are same max score targets
554                             max_score = max(target_scores.values())
555                             max_score_target_ids = [target_id for target_id,
    ↪ score in target_scores.items() if score == max_score]
556                             max_score_targets = [enemy for enemy in
    ↪ obs_data.enemies if enemy.id in max_score_target_ids]
557                             if len(max_score_targets) > 1:
558                                 # Chose the closest target
559                                 best_target = min(max_score_targets,
    ↪ key=lambda x: x.distance)
560                             else:
561                                 best_target = max_score_targets[0]
562                             if best_target.can_attack:
563                                 return attack(best_target.id)
564                             else:
565                                 # Best target is not in shoot range, move to
    ↪ target
566                                 dx = best_target.position[0]
567                                 dy = best_target.position[1]
568                                 if best_target.distance >
    ↪ obs_data.own_shoot_range / obs_data.own_sight_range:
569                                     path_action = find_path(obs_data, dx,
    ↪ dy, target_type=best_target.unit_type.lower())
570                                     if path_action:
571                                         return path_action
572                                     elif attack_actions:
573                                         if obs_data.last_action in
    ↪ attack_actions:
574                                             return obs_data.last_action
575                                         attackable_enemies = [enemy for
    ↪ enemy in obs_data.enemies if enemy.can_attack]
576                                         closest_enemy = min(
577                                             [enemy for enemy in
    ↪ attackable_enemies],
578                                             key=lambda enemy: enemy.distance,
579                                             default=None,
580                                         )
581                                         if closest_enemy and
    ↪ closest_enemy.can_attack:
582                                             return attack(closest_enemy.id)
583                                         return random.choice(attack_actions)
584             # If there are no enemies
585             else:
586                 # If there are only allies
587                 if obs_data.allies:
588                     # Check if any melee ally
589                     melee_ally = [ally for ally in obs_data.allies if
    ↪ ally.unit_type.lower() in ['zealot', 'zergling', 'baneling']]
590                     if melee_ally:
591                         # Move to center of melee allies
592                         melee_ally_x = sum(ally.position[0] for ally in
    ↪ melee_ally) / len(melee_ally)
593                         melee_ally_y = sum(ally.position[1] for ally in
    ↪ melee_ally) / len(melee_ally)
594                         dx = melee_ally_x
595                         dy = melee_ally_y
596                         distance = (dx ** 2 + dy ** 2) ** 0.5
597                         if distance > 0.05:
```

```
598                              path_action = find_path(obs_data, dx, dy)
599                              if path_action:
600                                  return path_action
601                      else:
602                          # No melee allies, move to target ally
603                          ally_x = sum(ally.position[0] for ally in
    ↪ obs_data.allies) / len(obs_data.allies)
604                          ally_y = sum(ally.position[1] for ally in
    ↪ obs_data.allies) / len(obs_data.allies)
605                          distance = (ally_x ** 2 + ally_y ** 2) ** 0.5
606                          if distance > 0.05:
607                              dx = ally_x
608                              dy = ally_y
609                              path_action = find_path(obs_data, dx, dy)
610                              if path_action:
611                                  return path_action
612          return default_action(obs)
613
614      return control_logic()
```

Listing 2: Example Script of Skill Initialzation

## I  BROADER IMPACT

We believe that the proposed work enhances the capacity for intelligent decision-making in complex and dynamic environments, and can have a positive impact on real-world multi-agent applications such as robotics, traffic management, and resource allocation. However, it is essential to consider potential concerns such as the discrepancy between the simulated environment and the real world. Another potential effect of directly implementing the derived policy is that it could lead to biased decision-making and privacy infringements. Mitigation strategies to address potential hazards could include the establishment of ethical guidelines and regulatory frameworks alongside the integration of transparency and explainability.

## J  THE USE OF LARGE LANGUAGE MODELS (LLMS)

Large Language Models (LLMs) served two distinct roles in this work. First, we used large language models as a light writing aid to improve grammar, phrasing, and formatting. The models were *not* used to propose ideas, design algorithms, write code, run experiments, analyze data, or draft scientific content. Second, and more centrally, Vision-Language Models (VLMs) are a fundamental component of our proposed COMPASS framework. As detailed throughout the manuscript, particularly in Section 4, VLMs are the core technology enabling the agents' perception, planning, and decision-making capabilities.

