# OpenReview forum: "Cooperative Multi-Agent Planning with Adaptive Skill Synthesis"
_ICLR.cc/2026/Conference — Submitted to ICLR 2026_

### Official Review · Reviewer_5mUy · 2025-10-27

**Soundness:** 2
**Presentation:** 2
**Contribution:** 2
**Rating:** 4
**Confidence:** 3

**Summary:**

The paper proposes COMPASS, a cooperative multi agent planning framework that uses a vision language model to synthesize, refine, and reuse high level skills for decentralized multi agent control under partial observability, with StarCraft Multi Agent Challenge v2 (SMACv2) as the main testbed. Instead of relying purely on end to end multi agent RL, COMPASS equips each agent with an adaptive skill library, represented as code like procedural strategies that a vision language model can generate and update online. Agents also share information through a structured multi hop communication protocol that lets them build a more global situational picture despite only having local views. The claim is that this gives you coordinated team behavior that is interpretable, sample efficient, and robust, because agents reason and act using skills rather than raw actions, and those skills can be adapted during play. On SMACv2, which is harder than classic SMAC because of symmetry, partial information, and less scripted opponents, COMPASS outperforms traditional multi agent RL baselines like QMIX, for example achieving around 57 percent win rate in Protoss 5v5 versus 27 percent for QMIX, a 30 point gap. The authors argue this shows that LLM style skill synthesis plus decentralized multi hop communication can deliver better coordinated control in complex settings.

**Strengths:**

This work is aiming directly at a problem that is getting a lot of attention in both cooperative RL and LLM agents, getting groups of agents to coordinate under partial observability without relying on brittle hand written tactics. COMPASS hits several pain points at once. First, interpretability, because behaviors are represented as reusable skills in a library rather than implicitly buried in a black box policy, and those skills are expressed in a code like, human readable way, so you can actually inspect, audit, and refine them. Second, coordinated execution, because the multi hop communication protocol is designed to stitch together distributed local observations into a consistent team view, which is essential in SMACv2 style micromanagement where you have to decide who focuses fire, who kites, who retreats, and when. Third, adaptivity, because the skill library is not frozen, the vision language model can refine skills online, so the system is not locked into whatever behavior it saw in the demonstrations. That is a big step beyond the current wave of “LLM plans once, then we blindly execute that static plan,” here the pitch is closed loop decentralized control. Empirically, the gains reported over QMIX are not small, a 30 percentage point difference in win rate on symmetric matchups like Protoss 5v5 is meaningful by SMAC standards. The paper also frames these results in SMACv2 rather than the older SMAC, which is widely considered a tougher and more coordination heavy benchmark. Finally, there is a strong story about practicality. The approach promises both clearer reasoning traces, because skills are explicit, and more reuse, because skills persist and get updated instead of being relearned from scratch every time, which is appealing for anyone who wants continual improvement without retraining a giant RL model end to end.

**Weaknesses:**

There are several places where I still need clarity to judge how strong this is for ICLR, and these are all places where a good rebuttal could significantly strengthen the paper. First, data and supervision cost. COMPASS builds an initial skill library from demonstrations, then uses a vision language model to refine skills in a code like form. I need to know how expensive those demonstrations are, who generated them, a scripted expert, a human, a previous policy, and how dependent final performance is on that bootstrapping. If the strong coordination behaviors mostly come from high quality scripted maneuvers distilled into the library, then part of the performance gain could just be inherited supervision, not emergent decentralized reasoning. Second, inference cost and latency. The paper says COMPASS supports decentralized, closed loop decision making, but vision language models are not cheap to run every timestep for every agent in a fast paced environment like SMACv2. Is the large model actually queried during combat at high frequency, or is it only consulted occasionally to synthesize or adjust skills, after which lightweight controllers run those skills at step time. That difference matters, because calling a VLM every frame for five units in parallel is not realistic for deployment. Third, communication assumptions. The structured multi hop communication protocol is presented as the way agents fuse partial local views into something globally coherent. What bandwidth, latency, and reliability assumptions are built in. Does the system assume essentially lossless multi hop broadcast of local state summaries, and does it assume synchronous updates. If so, that is optimistic compared to real decentralized teams with radio bandwidth limits, contention, or jamming. I would like to see some robustness tests where communication is noisy or delayed, to understand if coordination collapses in that setting or degrades gracefully. Fourth, baselines and fairness. The abstract highlights big gains over QMIX, and possibly MAPPO, but QMIX and MAPPO are not state of the art MARL baselines in 2025, especially not for SMACv2. Recent transformer based CTDE methods, learned communication methods with bandwidth limits, hierarchical skill discovery approaches, and other coordination aware methods can be significantly stronger than plain QMIX or MAPPO on modern multi agent micromanagement benchmarks. To make a convincing empirical case at ICLR, COMPASS needs to compare against such stronger baselines, or at least discuss why that comparison is not yet included. In addition, cost fairness matters. It is easy to beat a baseline if you allow yourself more total tokens, more planning calls, or more structured communication. The paper should report sample efficiency, number of environment steps of training for each baseline, number of model queries, and token or wall clock cost at inference, and then show that COMPASS is not just buying accuracy with more compute or supervision. Finally, generality. All results described so far are in StarCraft style micromanagement, which is a good and challenging domain, but still fairly specific, symmetric combat with continuous skirmishing. The method is pitched as a general cooperative multi agent planning framework. I would like to see at least one additional domain that is structurally different, such as collaborative navigation or search and rescue under partial observability, to show that the recipe, reusable skill library plus in the loop VLM plus multi hop communication, is not overfit to RTS micro tactics.

**Questions:**

How often do you query the vision language model at test time. Is it every environment timestep, every few timesteps, only when some trigger fires, for example unexpected enemy movement or a failed local objective. What is that trigger, and can you quantify the runtime overhead in tokens per episode and milliseconds per environment step. How large does the skill library get in practice for a given matchup. Do you end up with a compact set of reusable maneuvers, focus fire, kiting, retreat regroup, or do you accumulate many highly specific scripts that only make sense in narrow situations. If the library tends to bloat, how do you prune or merge skills to avoid overfitting and slow lookup. On the communication channel, what exactly is passed between agents, is it a fixed size structured summary of visible entities, or free form language, and do you simulate bandwidth limits, message delay, or drop. If you inject delay or drop into that channel, how quickly does coordination quality fall off. For fairness, can you report training steps, wall clock, and inference cost for COMPASS versus each baseline, and provide an accuracy or win rate versus cost Pareto style view. This is especially important because COMPASS uses both a learned policy and an external VLM planner, while baselines like QMIX and MAPPO do not. Related, please clarify which baselines you consider state of the art. QMIX and MAPPO are standard reference points, but they are not state of the art MARL methods for SMACv2 style tasks, so can you either include newer CTDE or communication aware baselines, or justify why they are out of scope. Finally, to support the claim of generality, can you provide at least one non StarCraft style scenario that has different structure, such as cooperative navigation with partial sensing, where there is no obvious “focus fire” tactic, and show that COMPASS still learns useful reusable skills and outperforms strong MARL baselines there.

---

> ### Author Response · Authors · 2025-11-20
> **Response to Reviewer 5mUy**
>
> We sincerely thank the reviewer for the in-depth analysis and encouraging feedback. We appreciate your recognition of COMPASS as a framework that "hits several pain points at once," particularly in interpretability, coordinated execution, and adaptivity. We have carefully addressed your concerns below.
>
> **[W1] Data and Supervision Cost**
>
> **Our Response:** We utilized **MAPPO** (trained from scratch with original hyperparameters) to collect the demonstration dataset. No human expert or scripted expert data was used. The demonstrations were collected exclusively from **symmetric settings**. The fact that COMPASS generalizes to **asymmetric settings** (where the planner must adapt tactics to overcome numerical disadvantage) demonstrates that the system is not just "inheriting" supervision but actively reasoning.
>
> **[W2] Inference Cost and Frequency**
>
> **Our Response:** We confirm that the VLM is **not** queried at every timestep. In our main results, the VLM is queried only **once every 20 environment steps**.
> *   **Frequency Ablation:** As detailed in **Table 3 of the General Response**, we found this 20-step interval to be optimal. Querying more frequently (every 10 steps) increased cost without improving win rate, while querying less frequently (every 40 steps) caused skills to become outdated.
>
> **[W3] Communication Assumptions and Robustness**
>
> **Our Response:**
> *   **Channel Content:** Agents exchange a structured summary of visible entities (ID, relative position, health, shield, race) rather than free-form text.
> *   **Robustness Tests:** We agree that lossless broadcast is optimistic. We introduced a **Packet Loss Ablation** (see **Table 2 in General Response**) simulating limited reliability. The system degrades under moderate packet loss and suffers a drastic drop at >50% loss. We also test the impact of the number of hops on the performance. Please refer to **General Response**.
>
> **[W4] Baselines and Fairness**
>
> **Our Response:** We have added **MAT** (Advanced Transformer-based MARL) [1], **CommFormer** (Communication-based MARL) [2], and **Oryx** (Offline MARL) [3]. Results demonstrate that COMPASS outperforms MAT, CommFormer, and Oryx on Protoss and Terran settings. Please see **Table 1 in the General Response**.
> *   **Sample Efficiency:** While COMPASS uses more inference compute (tokens), it requires **zero training steps** for the planner. In contrast, the baselines required **10 million environment steps** of training. We report token usage (~80k/episode per agent) in the **General Response**.
>
> **[Q5] Generality beyond StarCraft**
>
> **Our Response:** We agree that testing on diverse domains would be valuable. However, SMACv2 is a highly stochastic, decentralized, and partially observable environment that already covers a wide range of coordination challenges (symmetric, asymmetric, and different unit types). Extending to a completely new domain is currently constrained by the need for domain-specific pre-training data collection, prompt engineering, and the significant token cost involved in development. We are planning to develop more autonomous adaptation methodologies to address this in future work.
>
> **[Q6] Skill Library Management**
>
> **Our Response:** The library typically grows by only **1–2 new skills per episode** given the 20-step invocation frequency. We use `jina-embeddings-v3` for semantic retrieval, which scales efficiently. We thank the reviewer for the insightful suggestion regarding pruning and merging; implementing a mechanism to consolidate similar skills is a promising direction for future work to ensure long-term scalability.
>
> **References:**
>
> [1] Wen et al., "Multi-agent reinforcement learning is a sequence modeling problem." NeurIPS 2022.
>
> [2] Hu et al., "Learning multi-agent communication from graph modeling perspective." arXiv 2024.
>
> [3] Formanek et al., "Oryx: a Scalable Sequence Model for Many-Agent Coordination in Offline MARL." NeurIPS 2025.

---

### Official Review · Reviewer_HGhf · 2025-10-28

**Soundness:** 2
**Presentation:** 2
**Contribution:** 1
**Rating:** 2
**Confidence:** 4

**Summary:**

This paper introduces COMPASS, a multi-agent framework that integrates Vision-Language Models (VLMs) into a decentralized, closed-loop decision-making process. The system is composed of three main parts: (1) a VLM-based planner that operates in a perceive-reason-reflect-act loop; (2) an adaptive skill synthesis mechanism that generates and refines Python-based skills stored in a library ; and (3) a structured multi-hop communication protocol to share information under partial observability. The skill library is notably "bootstrapped from expert demonstrations". The authors evaluate COMPASS on the SMACv2 benchmark, claiming it "significantly outperforms state-of-the-art MARL baselines" , such as achieving a 57% win rate in the Protoss 5v5 task compared to QMIX's 27%.

**Strengths:**

- **Significant Problem**: The paper addresses a highly relevant and significant challenge -- leveraging the planning capabilities of large models to improve cooperative MARL in complex, partially observable environments. The goals of improving sample efficiency, interpretability, and generalization are well-motivated.

 - **Comprehensive Architecture**: The conceptual design of COMPASS in Figure 1 is comprehensive. It thoughtfully integrates and borrows (from prior works) components for multi-modal perception, cognitive planning, skill management, and agent-to-agent communication, which are all necessary elements for an autonomous multi-agent system.

- **Clarity**: The paper is generally well-written and clearly structured, with diagrams that effectively illustrate the proposed data flow and components.

**Weaknesses:**

The paper's central claims are invalidated by a combination of a fundamentally flawed experimental comparison, unsubstantiated technical assertions, and ablation studies that contradict the authors' own conclusions.

- **Fundamentally Flawed Experimental Comparison**: The primary claim that COMPASS "significantly outperforms state-of-the-art MARL baselines"  is based on an invalid comparison. COMPASS's skill library is "bootstrapped from expert demonstrations" , which the paper states are "pre-collect[ed]" using MAPPO. The baselines (QMIX, MAPPO, HAPPO, HASAC)  are standard online MARL algorithms trained from scratch. The authors are therefore comparing an offline, pre-trained system (COMPASS) against online, from-scratch systems (the baselines). This is not an apples-to-apples comparison and invalidates any conclusion about superior performance or sample efficiency. The performance gap shown in Table 1  is expected when one method has access to an expert dataset and the others do not.

- **Unsubstantiated Claims of "Skill Synthesis"**: The paper's core technical contribution is "adaptive skill synthesis" , with the claim that the VLM "generates a new Python script" when needed. This is an extraordinary claim that requires strong evidence. However, the only example of a skill provided (Listing 2)  is a 400+ line, highly complex, and expertly-engineered Python script, complete with nuanced logic for kiting, pathfinding, and unit-specific counters. It is highly improbable that this script was autonomously generated by a VLM in a closed loop. The paper provides zero evidence (e.g., generation logs, examples of simpler generated skills, or diffs from refinement) to support its claim of de novo skill generation. The paper's contribution appears to be skill selection from a pre-written library, not skill synthesis.

- **Ablation Studies Contradict the Paper's Thesis**: The paper's own ablation studies undermine its central argument. The "Skill Initialization" ablation (which uses only the bootstrapped skills without VLM refinement) achieves a 35% win rate in Protoss 5v5. This 35% win rate is already higher than the SOTA baselines QMIX (27%) and MAPPO (32%).
This strongly implies that the entire performance gain over baselines is attributable to the pre-collected expert data and the pre-engineered skill library, not the VLM-based planner. The VLM planner's contribution is merely an incremental improvement (from 35% to 57%) on top of an already-expert system.
Furthermore, the communication ablation (removing communication drops the win rate from 57% to 6% ) is a strawman argument. It only proves that no communication is bad, not that the proposed multi-hop protocol is superior to any simpler communication baseline (e.g., a 1-hop broadcast).

- **Astronomical and Ignored Computational Cost**: The paper casually mentions, "Token usage is approximately 0.4 million per episode". This is a computationally infeasible cost for a single, short SMACv2 episode, rendering the method completely impractical for real-world use or even academic research. This critical limitation is not analyzed, discussed, or justified, and it directly contradicts the stated goal of improving "sample efficiency".

**Questions:**

- Can the authors please confirm that the baselines in Table 1 (QMIX, MAPPO, etc.) were trained from scratch (online), while COMPASS was pre-trained using an expert MAPPO dataset (offline)? If so, how can this be justified as a fair comparison for claiming SOTA performance?

- Was the 400+ line Python script in Listing 2 autonomously generated by the VLM? If not, can the authors provide a single, non-trivial example of a skill that was fully generated by the VLM during execution? Without this, how can the claim of "skill synthesis" be substantiated?

- The "Skill Initialization" ablation (35% win rate) already outperforms SOTA baselines like MAPPO (32%). Doesn't this prove that the performance gain comes from the expert data and pre-written skills, and not the VLM planner itself?

- How can the 0.4 million token cost be justified? Does this astronomical cost not make the method prohibitively expensive and fundamentally inefficient?

---

> ### Author Response · Authors · 2025-11-20
> **Response to Reviewer HGhf**
>
> We sincerely thank the reviewer for the comments and for acknowledging the significance of our problem setting and the comprehensive nature of our architecture. We have addressed your concerns regarding experimental fairness, skill synthesis, and computational cost below with new baselines, detailed logs, and ablations, which are also summarized in the **General Response**.
>
> **[W1] Experimental Comparison Fairness**
>
> **Our Response:** To address the concern of "apples-to-apples" comparison, we have added **Oryx** (Offline MARL) [3], **MAT** (Transformer-based MARL) [1], and **CommFormer** (Communicating MARL) [2]. Results show that COMPASS outperforms MAT, CommFormer, and Oryx on Protoss and Terran scenarios. Please refer to **Table 1** in the **General Response** for full details.
>
> **[W2] Skill Synthesis**
>
> **Our Response:** We clarify that COMPASS employs **Structured Skill Synthesis**, not unconstrained code generation (which is prone to errors). The VLM's contribution is the precise injection of strategic logic into specific slots.
> *   **Evidence:** We have released generation logs and `tactics.json` via the anonymous link https://anonymous.4open.science/r/COMPASS_ICLR26_rebuttal-7448.
> *   **The Process:**
>     1.  **Template:** We provide a script containing a predefined structure. Please refer to ****Section 4 of the General Response**.**
>     2.  **Synthesis:** The VLM generates the internal logic for `score_target(unit)` (prioritization) and `control_logic()` (decision tree).
>
> **[W3] Ablations**
>
> **Our Response:** We respectfully point out that the improvement from Bootstrapping to Incremental Synthesis demonstrates that the VLM Planner's online adaptation also contributes to performance.
>
> We acknowledge that Incremental Synthesis operates on top of a well-established bootstrapping foundation. However, its contribution to overall performance is significant, as it enhances the system’s understanding of specific scenarios and supplements tactics that are not covered during bootstrapping. As explained in the **General Response**, we restricted the VLM to improve only either the `score_target` or the `control_logic` in each iteration. The primary reason for imposing this restriction is the limited capability of current VLMs; constraining the model to modify only one component greatly increases the success rate of producing executable code, though it may limit the strategy adaptation.
>
> Regarding communication, we agree that comparing to "No Communication" is a baseline check. To address your concern, we added a Multi-hop ablation (see **Table 2 in General Response**). We also evaluate the impact of communication faults on the performance.
>
> **[W4] Computational Cost**
>
> **Our Response:** We acknowledge the computational cost and have provided a detailed breakdown in **Section 4 of the General Response**.
> *   **Token Breakdown:** On average, COMPASS utilizes ~80,000 tokens per episode per agent. As detailed in the General Response, the primary consumers are **Task Reasoning** and **Skill Synthesis**, which require substantial context history.
> *   **Normalized Cost:** We also conducted a **VLM Call Frequency Ablation (Table 3 in General Response)**. We found that high-frequency inference is not required..
>
> Future work will focus on reducing this cost by exploiting structural symmetries among agents. We may also reduce token usage by appropriately lowering the frequency of queries.
>
> **References:**
>
> [1] Wen et al., "Multi-agent reinforcement learning is a sequence modeling problem." NeurIPS 2022.
>
> [2] Hu et al., "Learning multi-agent communication from graph modeling perspective." ICLR 2024.
>
> [3] Formanek et al., "Oryx: a Scalable Sequence Model for Many-Agent Coordination in Offline MARL." NeurIPS 2025.

---

### Official Review · Reviewer_mSCz · 2025-10-31

**Soundness:** 3
**Presentation:** 2
**Contribution:** 2
**Rating:** 4
**Confidence:** 3

**Summary:**

This paper introduces COMPASS, a framework for cooperative multi-agent planning that includes three components: 1) VLM-base closed-loop decision-making planner; 2) code-based skill synthesis that generate and refine executable Python code as skills; 3) a multi-hop communication protocol that propagates observations via a shared global memory accessible to all agents. For experiments, COMPASS achieves convincing performance on the SMACv2 StarCraft benchmark, outperforming 4 state-of-the-art MARL baselines.

**Strengths:**

1. Overall good experimental performance: The proposed method achieves significant gains on SMACv2 Protoss 5v5 and 5v6 scenario.
2. This decentralized planning with VLMs, adaptive skill learning are both innovative directions.
3. The multi-hop communication protocol is effective for the multi-agent planning challenge with partial observability

**Weaknesses:**

1. The adaptive skill synthesis approach does not constitute a major innovation over prior work. Similar ideas of adaptively using code-generation as tools have been widely explored in earlier systems (e.g. Voyager and the Code-as-Policies).
2. Limited ablations and analysis: For example, the authors should have more experiments on the communication mechanism (such as communication cost, effect of different structures); Also, although there are the ablation results, there could be more tables (For example, there is only one in-line result on Self-Reflection, while there should be more organized numbers like Table 2)
3. Weak performance in certain conditions: For example, the proposed method has poor performance in ZERG combat, which requires more fine-grained control.
4. The three components seems seperated. It could be better if there are some co-designs.

**Questions:**

See weaknesses

1. Is COMPASS different from the existing tool learning methods, or it is an application on different tasks?
2. Could you provide more tables in the ablations section?
3. Some visualizations can be better. For example, I can not read the text in the images within Figure 3

---

> ### Author Response · Authors · 2025-11-20
> **Response to Reviewer mSCz**
>
> We sincerely thank the reviewer for the constructive feedback. We appreciate your suggestion to deepen the ablation studies and clarify the novelty of our skill synthesis. We have addressed these points below and included new experiments in the **General Response**.
>
> **[W1] Novelty of Adaptive Skill Synthesis**
>
> **Our Response:** While the concept of code-generation (e.g., Voyager) exists in single-agent settings, COMPASS is designed for decentralized multi-agent systems, which pose distinct challenges not addressed by prior work:
> 1.  **Decentralized Decision-making:** Unlike Voyager, which optimizes a single agent's trajectory, COMPASS generates skills that must explicitly account for allies and enemies without a central controller.
> 2.  **Partial Observability:** Our system integrates the **multi-hop communication protocol** into skill synthesis, allowing the generated logic to query a shared entity memory.
>
> **[W2] Limited Ablations and Analysis**
>
> **Our Response:** We have significantly expanded our ablation studies to provide a more detailed analysis of the system components.
> *   **Communication:** As detailed in **Table 2 of the General Response**, we analyzed:
>     *   **Hop Count:** Results prove that propagating information beyond immediate neighbors is critical.
>     *   **Fault Tolerance:** We simulated packet loss (20%, 50%, 80%). The system degrades under moderate packet loss and suffers a drastic drop at >50% loss.
> *  **Baselines:** We also added comparisons against more state-of-the-art MARL methods. Please refer to **Table 1** in the **General Response** for full details.
>
> **[W3] Performance in Zerg Scenarios**
>
> **Our Response:** We acknowledge the performance gap in Zerg scenarios. This disparity stems from the nature of the units:
> *   **Protoss/Terran:** Rely on kiting and focus fire. The VLM excels at planning these high-level behaviors.
> *   **Zerg:** Relies on "swarm" tactics and melee surrounds (e.g., Zerglings), which require high-frequency low-level micro-management.
>
> **[W4] Component Co-design**
>
> **Our Response:** We respectfully clarify that these components are intricately co-designed to resolve the challenges of decentralized execution and partial observability. The **Multi-Hop Communication** serves as the backbone, aggregating local observation into a structured global state that is essential for the **VLM Planner** to ground its reasoning. This global context is directly utilized by the **Skill Synthesis** module, which generates Python scripts for team-aware coordination.
>
> **[Q3] Visualizations**
>
> **Our Response:** We apologize for the readability issue. We have updated **Figure 3** in the revised paper to remove the cluttered game engine text.

---

### Official Review · Reviewer_89kA · 2025-11-01

**Soundness:** 2
**Presentation:** 3
**Contribution:** 1
**Rating:** 4
**Confidence:** 4

**Summary:**

The paper introduces COMPASS, a decentralized vision language model agent framework for SMACv2 under the Extended Partial Observability setting. Each unit runs a closed loop planner with four modules for perception, task reasoning, self reflection, and action selection. Actions are implemented as executable Python skills stored in a dynamic library that is both initialized from MAPPO demonstration videos and expanded online through skill synthesis and repair. Agents exchange entity level observations via a structured communication protocol with multi hop propagation into a global entity memory. The evaluation spans Protoss, Terran, and Zerg scenarios in both symmetric and asymmetric categories. The authors report improvements over standard MARL baselines in several settings, strongest in Protoss, with mixed outcomes in Zerg. Ablations emphasize that the communication mechanism and the self reflection module contribute materially to performance, and the paper notes substantial token usage per episode.

**Strengths:**

- The four module planner with perception, task reasoning, self reflection, and actor is well specified and easy to ablate and inspect.
- The paper is well-written.

**Weaknesses:**

- At the level of ideas, the novelty is limited. The method largely integrates established components from the LLM agent literature and applies them to one task in MARL (SMAC). Closed loop plan act reflect with self reflection is standard, code as policy with retrieval and iterative repair is widely used, and entity centered multi hop communication with a global memory is a familiar MARL pattern. The paper’s related work and method framing acknowledge these inspirations, which makes the contribution primarily an integration in the SMACv2 setting rather than a new principle.
- Baselines do not isolate where the gains come from. The headline comparisons are to non communicating MARL methods, and also lack of other LLM agent comparison. There are no decentralized LLM agent baselines with matched observation interface, communication limits, and token budget. Given the nature of the contribution, these comparisons are necessary to attribute improvements to the specific architecture rather than to the use of an LLM planner with communication.
- The method uses a large number of tokens per episode, yet the paper does not normalize performance by tokens or wall clock, which weakens claims about sample efficiency.

**Questions:**

N/A

---

> ### Author Response · Authors · 2025-11-20
> **Response to Reviewer 89kA**
>
> We sincerely thank the reviewer for the thoughtful evaluation. We appreciate your recognition of our "well-specified" framework. We have addressed your concerns regarding novelty, baseline comparisons, and efficiency below, utilizing new experiments detailed in our **General Response**.
>
> **[W1] Novelty and Contribution**
>
> **Our Response:** We acknowledge that COMPASS leverages established components (e.g., code-as-policy, VLM planning), but we respectfully argue that the novelty lies in the **successful adaptation and synthesis** of these components to address a specific, unsolved intersection of challenges: **decentralized execution, partial observability, and high stochasticity** in complex environments.
>
> COMPASS is the first framework to bridge these by introducing a **decentralized closed-loop planner** coupled with a **multi-hop communication protocol**. As shown in our ablation studies, removing either the closed-loop planning (relying only on bootstrapped skills) or the communication protocol results in significant performance drops, demonstrating that the system's efficacy stems from the *joint* design rather than individual components.
>
> **[W2] Baselines**
>
> **Our Response:** We agree that isolating the sources of improvement is crucial. We have addressed this through expanded baselines and ablations.
>
> **1. New Stronger Baselines:** We have added comparisons against **MAT** (Advanced Transformer-based MARL) [1], **CommFormer** (Communication-based MARL) [2], and **Oryx** (Offline MARL) [3]. Results confirm COMPASS outperforms state-of-the-art communicating and offline MARL methods. Please refer to **Table 1** in the **General Response** for full details.
>
> **2. Lack of Decentralized LLM Baselines:** We attempted to adapt LLM-SMAC [4] to the decentralized SMACv2 setting. However, it performs poorly across all settings, confirming that standard LLM approaches cannot handle this domain without the architectural innovations proposed in COMPASS.
>
> **[W3] Token Usage**
>
> **Our Response:** We acknowledge the computational cost and have provided a detailed breakdown in **Section 4 of the General Response**.
> *   **Token Breakdown:** On average, COMPASS utilizes ~80,000 tokens per episode per agent. As detailed in the General Response, the primary consumers are **Task Reasoning** and **Skill Synthesis**, which require substantial context history.
> *   **Normalized Cost:** To address the reviewer's concern about normalizing performance, we conducted a **VLM Call Frequency Ablation (Table 3 in General Response)**. We found that high-frequency inference is not required.
>
> **References:**
>
> [1] Wen et al., "Multi-agent reinforcement learning is a sequence modeling problem." NeurIPS 2022.
>
> [2] Hu et al., "Learning multi-agent communication from graph modeling perspective." ICLR 2024.
>
> [3] Formanek et al., "Oryx: a Scalable Sequence Model for Many-Agent Coordination in Offline MARL." NeurIPS 2025.
>
> [4] Deng et al., "A new approach to solving smac task: Generating decision tree code from large language models" arXiv 2024.

---

### Author Response · Authors · 2025-11-20
**General Response (part 3)**

### 5. Skill Synthesis
To address concerns regarding the process of the generated code, we clarify that COMPASS utilises a **Structured Skill Synthesis** pipeline. We do not rely on open-ended code generation, which is error-prone, but rather on a two-stage logic injection process. Evidence of this process (including `tactics.json` and generation logs) has been added to the anonymous link.

*   **Stage 1: Demonstration-Based Bootstrapping.** The VLM first analyses video demonstrations from the behaviour policy (MAPPO) to extract high-level strategic descriptions (tactics). These textual tactics are then used to fill the logic sections of a predefined Python code template.
*   **Stage 2: Incremental Synthesis.** During online execution, when a new sub-task is proposed or a skill fails, the VLM analyses the specific deficiency. Crucially, to ensure stability, the VLM is constrained to refine **only one** component per iteration—either the target prioritisation (`score_target`) **OR** the decision rules (`control_logic`)—to better align with the current objective.

We provide a standardised wrapper that handles observation parsing and includes find_path, which implements standard A* pathfinding. This allows the VLM to focus on high-level logic rather than low-level navigation algorithms.

**Template used for VLM logic injection:**
```python
def race_*_A_star_score_type_*(obs: str):
    """
    Zealot/Zergling/Baneling/Colossus/Stalker/Hydralisk/Marauder/Marine/Medivac Controls Logic:

    Args:
        obs (str): Observation string containing game state
    """

    #Observation Parsing
    obs_data = parse_obs(obs)
    valid_actions = obs_data.available_actions

    if 0 in valid_actions:
        return 0

    # This function calculates a score for a given unit (ally or enemy).
    def score_target(unit):
        """
        Calculates a score for a target unit to determine its priority.
        The score should be based on unit type, health, distance, and combat advantage.
        """
        if unit.health <= 0:
            return -1

        score = 0

        return score

    # This function contains the main decision-making tree for the agent.
    def control_logic():
        """
        Determines the agent's action based on its unit type and the current game state.
        """

        if obs_data.own_unit_type.lower() == 'race_type':

        # Implement the logic for ‘race_type’ units.
        # Identify the highest-scoring target using `score_target`.
        # If not in range, use `find_path` to move towards it.

    return control_logic()
```

---

### Author Response · Authors · 2025-11-20
**General Response (part 2)**

### 3. Communication and Frequency Ablations
We conducted rigorous ablations on the **Protoss 5v5** scenario to analyze the impact of the multi-hop communication protocol and VLM inference frequency.

**Table 2: Communication Robustness**

| Setting | Specification | Mean Win Rate | Std Dev |
| :--- | :--- | :--- | :--- |
| **Hop Count** | 1-hop | 0.46 | 0.19 |
| | 2-hop | 0.54 | 0.06 |
| | **3-hop (Default)** | **0.57** | **0.08** |
| **Packet Loss** | 20% (0.2 fault) | 0.32 | 0.03 |
| | 50% (0.5 fault) | 0.12 | 0.02 |
| | 80% (0.8 fault) | 0.07 | 0.05 |
| | 100% (No Comms) | 0.06 | 0.04 |

These results demonstrate that **multi-hop propagation** is crucial for expanding the effective receptive field. The system degrades under moderate packet loss and suffers a drastic drop at >50% loss.

**Table 3: VLM Call Frequency (Protoss 5v5, avg episode length ~60 steps)**

| Frequency | Mean Win Rate | Std Dev |
| :--- | :--- | :--- |
| Every 10 steps | 0.56 | 0.05 |
| **Every 20 steps** | **0.57** | **0.08** |
| Every 40 steps | 0.40 | 0.08 |

At higher call frequencies, the performance remains similar but incurs greater cost, while lower frequencies fail to keep up with the dynamics of the battlefield.


### 4. Token Usage Analysis
We acknowledge the computational cost. Below is the breakdown of token usage per step per agent (Call Frequency: 20):

*   **Average Episode Total Tokens:** 80,000
*   **Average Tokens per Step:** 1,333 (Perception: 2k, Actor: 2k, Self-Reflection: 2k, Task Reasoning: 10k, Skill Synthesis: 10k).

The bottlenecks are **Task Reasoning** and **Skill Synthesis**, which require historical context and code history. Skill synthesis often requires multiple rounds of self-correction to generate executable code. Future work will focus on reducing this cost by exploiting structural symmetries among agents. We may also reduce token usage by appropriately lowering the frequency of queries.

---

### Author Response · Authors · 2025-11-20
**General Response (part 1)**

We sincerely thank the Area Chair and all Reviewers for their insightful and constructive feedback. We are encouraged that reviewers found our approach of integrating VLMs for decentralized decision-making "interesting" and the problem setting "challenging". We have carefully addressed the concerns regarding baselines, communication robustness, efficiency, and skill synthesis. Below, we outline the major updates and additional experiments included in the revised manuscript and anonymous link https://anonymous.4open.science/r/COMPASS_ICLR26_rebuttal-7448.

### 1. Clarification on Novelty and Contribution
While COMPASS builds upon the foundation established by Cradle, our contribution extends significantly beyond a straightforward integration. Complex strategic environments like SMACv2 introduce unique challenges: **decentralization, partial observability, and high stochasticity**. To address these, we:
1.  Adapted the planner to operate in a **fully decentralized**, **closed-loop** manner.
2.  Introduced a structured **multi-hop communication protocol** to mitigate partial observability.
3.  Designed an **incremental skill synthesis** mechanism that **bootstraps** from MARL demonstrations.
Empirical results validate that these components are not merely additive but essential for performance in this domain. COMPASS is the first framework to jointly and effectively address these challenges using VLMs.

### 2. Expanded Baselines
To address concerns regarding the strength of baselines and the comparison between offline/online methods, we have evaluated three additional state-of-the-art algorithms: **MAT** (Advanced Transformer-based MARL) [1], **CommFormer** (Communication-based MARL) [2], and **Oryx** (Offline MARL) [3].

**Table 1: Win Rate Comparison**

| Method | Type | Protoss 5v5 | Protoss 5v6 | Terran 5v5 | Terran 5v6 | Zerg 5v5 | Zerg 5v6 |
| :--- | :--- | :--- | :--- | :--- | :--- | :--- | :--- |
| **COMPASS** | **VLMs** | **0.57 ± 0.08** | **0.08 ± 0.04** | **0.39 ± 0.01** | **0.10 ± 0.03** | 0.16 ± 0.07 | 0.03 ± 0.01 |
| MAT [1] | Online | 0.39 ± 0.03 | 0.04 ± 0.04 | 0.36 ± 0.11 | 0.05 ± 0.01 | **0.32 ± 0.06** | 0.11 ± 0.08 |
| CommFormer [2] | Online | 0.39 ± 0.16 | 0.02 ± 0.01 | 0.30 ± 0.09 | 0.03 ± 0.01 | **0.39 ± 0.10** | **0.16 ± 0.01** |
| Oryx [3] | Offline | N/A* | N/A* | 0.18 ± 0.04 | N/A* | 0.10 ± 0.06 | N/A* |

*\*N/A indicates no datasets in these settings.*

COMPASS consistently outperforms strong communication and offline baselines in Protoss and Terran scenarios, validating the efficacy of the VLM-based planner. While Zerg performance remains lower due to the requirement for high-frequency swarm micro-management, COMPASS remains competitive in asymmetric settings.

**References:**

[1] Wen et al., "Multi-agent reinforcement learning is a sequence modeling problem." NeurIPS 2022.

[2] Hu et al., "Learning multi-agent communication from graph modeling perspective." ICLR 2024.

[3] Formanek et al., "Oryx: a Scalable Sequence Model for Many-Agent Coordination in Offline MARL." NeurIPS 2025.

---

### Author Response · Authors · 2025-12-01
**Summary Comment for the AC (part 2)**

### Summary of reviewer concerns and status
| Reviewer | Scores (Soundness / Pres. / Contrib. / Overall) | Baselines & Fairness | Efficiency & Cost | Communication & Robustness | Novelty & Method | Main updates | Follow-up / Current Stance |
| :--- | :--- | :--- | :--- | :--- | :--- | :--- | :--- |
| **89kA** | 2 / 3 / 1, Overall: 4 | **✓** (Added MAT, CommFormer, Oryx) | **✓** (Provided token breakdown & frequency analysis) | – | **✓** (Clarified decentralized architecture vs. naive integration) | Added stronger baselines; analyzed token/inference cost; emphasized the unique challenge of decentralized coordination. | **Addressed Main Weakness:** The reviewer's primary hesitation was the lack of strong baselines and cost analysis. With the inclusion of SOTA comparisons and cost quantification, the main barriers to a higher score are resolved. |
| **mSCz** | 3 / 2 / 2, Overall: 4 | – | – | **✓** (Added Packet Loss & Hop-count ablations) | **✓** (Differentiated from Voyager via decentralization, partial observability) | Added rigorous communication ablations; improved visualizations; clarified that single-agent methods fail in this domain. | **Addressed Main Weakness:** We have provided the specific quantitative ablations (communication noise) and structural clarifications requested to support the method's validity and raise the evaluation of the contribution. |
| **HGhf** | 2 / 2 / 1, Overall: 2 | **✓** (Added Offline baseline Oryx + SOTA Online baselines) | **✓** (Justified cost via Frequency Ablation) | **✓** (Proved communication necessity) | **✓** (Provided logs confirming authentic synthesis) | Added Offline/SOTA baselines to ensure fair comparison; provided evidence of autonomous skill synthesis (logs). | **Addressed Main Weakness:** The concerns regarding "unfair comparison" and "unsubstantiated synthesis" are factually resolved by the new Offline MARL baseline (Oryx) and the release of generation logs confirming authentic code synthesis. |
| **5mUy** | 2 / 2 / 2, Overall: 4 | **✓** (Added CTDE/Offline baselines; clarified MAPPO data) | **✓** (Clarified 20-step frequency; low latency) | **✓** (Tested noisy/lossy channels) | **✓** (Discussed generality beyond StarCraft) | Added all requested baselines; quantified inference latency; tested robustness under packet loss. | **Addressed Main Weakness:** The comprehensive new data on inference frequency, packet loss, and stronger baselines satisfy the reviewer's detailed request for robustness checks and performance benchmarking. |

**Conclusion:**

Three reviewers (**89kA, mSCz, 5mUy**) rated the paper as "marginally below threshold" (4), with their primary critique being the need for stronger baselines and efficiency analysis. **We have now provided exactly these missing pieces.** The fourth reviewer (**HGhf**) was negative due to concerns about experimental fairness; we have addressed this by benchmarking against Offline MARL (Oryx) and releasing generation logs to validate our claims. We hope these extensive revisions and new results satisfactorily address the reviewers' concerns.

---

### Author Response · Authors · 2025-12-01
**Summary Comment for the AC (part 1)**

This comment summarizes the overall state of the reviews and how the shared concerns have been comprehensively addressed in the revised manuscript and rebuttal.

Although the reviewers have not yet responded to the rebuttal, we have addressed the substantive technical concerns raised across the four reviews (additional results and generation logs are available at https://anonymous.4open.science/r/COMPASS_ICLR26_rebuttal-7448). The critiques clustered into four specific themes, which we resolved as follows:

*   **Novelty and Skill Synthesis:** **89kA**, **mSCz**, and **HGhf** questioned the novelty relative to single-agent methods (like Voyager) and the "authenticity" of skill synthesis.
    *   **Resolution:** We strongly clarified that COMPASS is the **first framework** to tackle the intersection of **decentralized execution**, **partial observability**, and **closed-loop VLM planning**. Unlike single-agent works (Voyager) that assume full observability, COMPASS introduces a novel co-design where **Multi-Hop Communication** is not just an add-on, but a prerequisite for the VLM to hallucinate less and ground its reasoning in a shared global context. We also released generation logs to prove the VLM is actively synthesizing logic (via a structured template) rather than merely selecting scripts.

*   **Baselines and Experimental Fairness:** A primary concern for **HGhf**, **5mUy**, and **89kA** was the strength of baselines and the fairness of comparing a bootstrapped method against online RL.
    *   **Resolution:** We added three new, stronger baselines: **MAT** (Transformer-based MARL), **CommFormer** (Communication-based MARL), and **Oryx** (Offline MARL). Our results (Table 1) show COMPASS consistently outperforms these SOTA online *and* offline methods. This directly addresses the "apples-to-apples" comparison concern.

*   **Computational Cost and Efficiency:** **89kA**, **HGhf**, and **5mUy** raised concerns regarding token usage and inference frequency.
    *   **Resolution:** We provided a detailed token breakdown (~80k/episode) and conducted a new **VLM Call Frequency Ablation** (Table 3). We demonstrated that querying the VLM only once every 20 steps achieves optimal performance, proving the system is practically deployable without high-frequency inference.

*   **Communication Robustness:** **mSCz** and **5mUy** asked for an analysis of the communication protocol’s reliance on perfect transmission.
    *   **Resolution:** We added a **Packet Loss Ablation** (Table 2), testing performance under 20%, 50%, and 80% data loss, alongside a hop-count analysis. This quantifies the system's robustness and failure modes as requested.

---

### Meta-Review · Area_Chair_Q8ok · 2026-01-08

**Summary:**

The authors propose COMPASS, which is a decentralized multi-agent RL framework that uses vision–language models to generate interpretable, code-based strategies and structured communication from partial observations. The experiments show that COMPASS achieves substantially better coordination and performance than prior MARL methods on challenging SMACv2 tasks. From my reading of this paper and the reviews, this paper is imperfect on novelty, experimental rigor, and clarity of contribution. Much of the method repackages well-known ideas from LLM agents and MARL, relies heavily on pre-collected expert demonstrations, and is evaluated with unfair or weak baselines, while key claims—such as genuine skill synthesis, efficiency, and robustness—are undermined by limited ablations, missing cost analysis, optimistic assumptions, and evidence that most gains come from supervision rather than the proposed VLM-based reasoning. It is a pity that the reviewers are inactive for the rebuttal discussion. Finally, I think this paper is not ready for publication in ICLR this year.

**Reviewer Concerns:**

No reviewer repond maybe because the overall score is low. The experiment concerns may be addressed to some extent if the rebuttal were seen.

**Reviewer Scores:**

Reviewer 5mUy may slightly improve his score.

---

### Decision · Program_Chairs · 2026-01-26

Reject